# Hypocretin underlies the evolution of sleep loss in the Mexican cavefish

James B Jaggard[1], Bethany A Stahl[1], Evan Lloyd[1], David A Prober[2], Erik R Duboue[3,4], Alex C Keene[1]*

[1]Department of Biological Sciences, Florida Atlantic University, Jupiter, United States; [2]Division of Biology and Biological Engineering, California Institute of Technology, Pasadena, United States; [3]Department of Embryology, Carnegie Institution for Science, Baltimore, United States; [4]Harriet L. Wilkes Honors College, Florida Atlantic University, Jupiter, United States

**Abstract** The duration of sleep varies dramatically between species, yet little is known about the genetic basis or evolutionary factors driving this variation in behavior. The Mexican cavefish, *Astyanax mexicanus*, exists as surface populations that inhabit rivers, and multiple cave populations with convergent evolution on sleep loss. The number of Hypocretin/Orexin (HCRT)-positive hypothalamic neurons is increased significantly in cavefish, and HCRT is upregulated at both the transcript and protein levels. Pharmacological or genetic inhibition of HCRT signaling increases sleep in cavefish, suggesting enhanced HCRT signaling underlies the evolution of sleep loss. Ablation of the lateral line or starvation, manipulations that selectively promote sleep in cavefish, inhibit *hcrt* expression in cavefish while having little effect on surface fish. These findings provide the first evidence of genetic and neuronal changes that contribute to the evolution of sleep loss, and support a conserved role for HCRT in sleep regulation.
DOI: https://doi.org/10.7554/eLife.32637.001

*For correspondence:
keenea@fau.edu

**Competing interests:** The authors declare that no competing interests exist.

## Introduction

Sleep behavior is nearly ubiquitous throughout the animal kingdom and is vital for many aspects of biological function (*Campbell and Tobler, 1984*; *Hartmann, 1973*; *Musiek et al., 2015*). While animals display remarkable diversity in sleep duration and architecture, little is known about the functional and evolutionary principles underlying these differences (*Allada and Siegel, 2008*; *Capellini et al., 2008*; *Siegel, 2005*). We previously discovered the convergent evolution of sleep loss in the blind Mexican cavefish, *Astyanax mexicanus* (*Duboué et al., 2011*). Relative to extant conspecifics that inhabit caves, independently derived cave-dwelling populations display a striking 80% reduction in total sleep with no adverse impacts on health or development (*Duboué et al., 2011*). The robust differences in sleep between surface and cave populations provide a unique model for investigating the genetic basis for sleep variation and identification of novel mechanisms underlying the evolution of sleep regulation.

*Astyanax mexicanus* consists of eyed surface populations that inhabit rivers in the Sierra del Abra region of Northeast Mexico, and at least 29 distinct populations of cavefish in this region (*Mitchell et al., 1977*). Cavefish are derived from surface ancestors, which arose from colonization events that are estimated to have taken place within the past 2–5 million years (*Gross, 2012*; *Jeffery, 2009*; *Keene et al., 2015*). Independently-evolved cave populations of *A. mexicanus* share morphological and developmental phenotypes including smaller or completely absent eyes, and loss of pigmentation (*Borowsky, 2008a*; *Gross and Wilkens, 2013*; *Protas et al., 2006*). In addition, cavefish display an array of behavioral changes including reduced schooling, enhanced vibration attraction behavior, hyperphagia, and sleep loss (*Aspiras et al., 2015*; *Duboué et al., 2011*;

*Kowalko et al., 2013*; *Yoshizawa et al., 2010*). Convergent evolution of shared traits in independent cavefish populations, combined with robust phenotypic differences with extant surface fish populations, provides a system to examine how naturally occurring variation and evolution shape complex biological traits.

While the ecological factors underlying phenotypic changes in cave populations are unclear, food availability and foraging strategy are hypothesized to be potent drivers of evolutionary change that contribute to the variation in sleep duration across animal species (*Siegel, 2005*). Many cave waters inhabited by *A. mexicanus* are nutrient poor compared to the above-ground rivers surrounding them (*Mitchell et al., 1977*), and previous field studies suggest cavefish subsist primarily off of bat guano, small insects, and organic matter washed into the cave by seasonal floods (*Keene et al., 2015*; *Mitchell et al., 1977*). Following starvation, cave-derived fish have a slower rate of weight loss compared to surface conspecific, suggesting that a reduced metabolism may account, in part, for adaptation to cave life (*Aspiras et al., 2015*). We previously found that sleep is increased in cavefish during periods of prolonged starvation, raising the possibility that cavefish suppress sleep to forage during the wet season when food is plentiful, and increase sleep to conserve energy during the dry season when food is less abundant (*Jaggard et al., 2017*). Therefore, sleep loss in cavefish appears to be an evolved consequence of changes in food availability, providing a model to examine interactions between sleep and metabolism.

Despite the robust phenotypic differences in sleep between *A. mexicanus* surface and cave populations, little is known about the neural mechanisms underlying the evolution of sleep loss in cavefish. Many behaviors that are altered in cavefish are regulated by the hypothalamus, which is enlarged in cavefish (*Menuet et al., 2007*). Here, we investigate the role of Hypocretin/Orexin (HCRT), a highly conserved hypothalamic neuropeptide known to consolidate wakefulness. Deficiencies in HCRT signaling are associated with altered sleep and narcolepsy-associated phenotypes in diverse vertebrate organisms (*Appelbaum et al., 2009*; *Chemelli et al., 1999*; *Faraco et al., 2006*; *Lin et al., 1999*; *Prober et al., 2006*; *Yokogawa et al., 2007*). In zebrafish, HCRT is critical for normal sleep-wake regulation. Ectopic expression of *hcrt* increases locomotor activity, while ablation of HCRT neurons increases daytime sleep (*Elbaz et al., 2012*; *Prober et al., 2006*; *Singh et al., 2015*). We now show that HCRT expression is down-regulated in cavefish in response to sleep-promoting manipulations including starvation and ablation of the lateral line (*Jaggard et al., 2017*). Further, pharmacologic or genetic disruption of HCRT signaling selectively restores sleep to cavefish but not surface fish. Together, these findings suggest plasticity of HCRT function contributes to evolved differences in sleep regulation in Mexican cavefish.

## Results

Sleep is dramatically reduced in adult Pachón cavefish compared to surface fish counterparts (*Figure 1A,B*) (*Jaggard et al., 2017*; *Yoshizawa et al., 2015*). We compared sequence homology between surface fish and cavefish by a bioinformatic analysis of the sequences from the cavefish genome (*McGaugh et al., 2014*) and available full-length transcriptomic sequences (*Gross et al., 2013*). Alignment of the HCRT neuropeptide reveals that *A. mexicanus* shares high sequence similarity to other fish species (35–48% percent identity) and mammals (35% percent identity), including conservation of domains that give rise to the HCRT neuropeptides (*Figure 1—figure supplement 1A*). The HCRT peptide sequences of surface and Pachón cavefish are identical (100% percent identity). To determine if *hcrt* expression is altered in adult cavefish, we measured transcript levels in whole-brain extracts with quantitative real-time PCR (qPCR). Expression of the housekeeping gene, GAPDH, was comparable between both forms (*Figure 1—figure supplement 1B*). By contrast, *hcrt* expression was significantly elevated in Pachón cavefish to over three-fold the levels of surface fish, raising the possibility that upregulation of *hcrt* underlies the evolution of sleep loss (*Figure 1C*). Neuropeptide levels were quantified by immunolabeling serial-sectioned brains, and examining the number of HCRT-positive cell bodies and the relative fluorescence of each cell under fed conditions (*Figure 1—figure supplement 2A–D*). The number of HCRT-positive cell bodies was significantly higher in Pachón cavefish compared to surface fish (*Figure 1D*). Further, quantification of fluorescence intensity of individual cells revealed increased HCRT neuropeptide in cavefish (*Figure 1E–I*). Enhanced levels of HCRT protein were also observed in five day post fertilization (dpf) larvae,

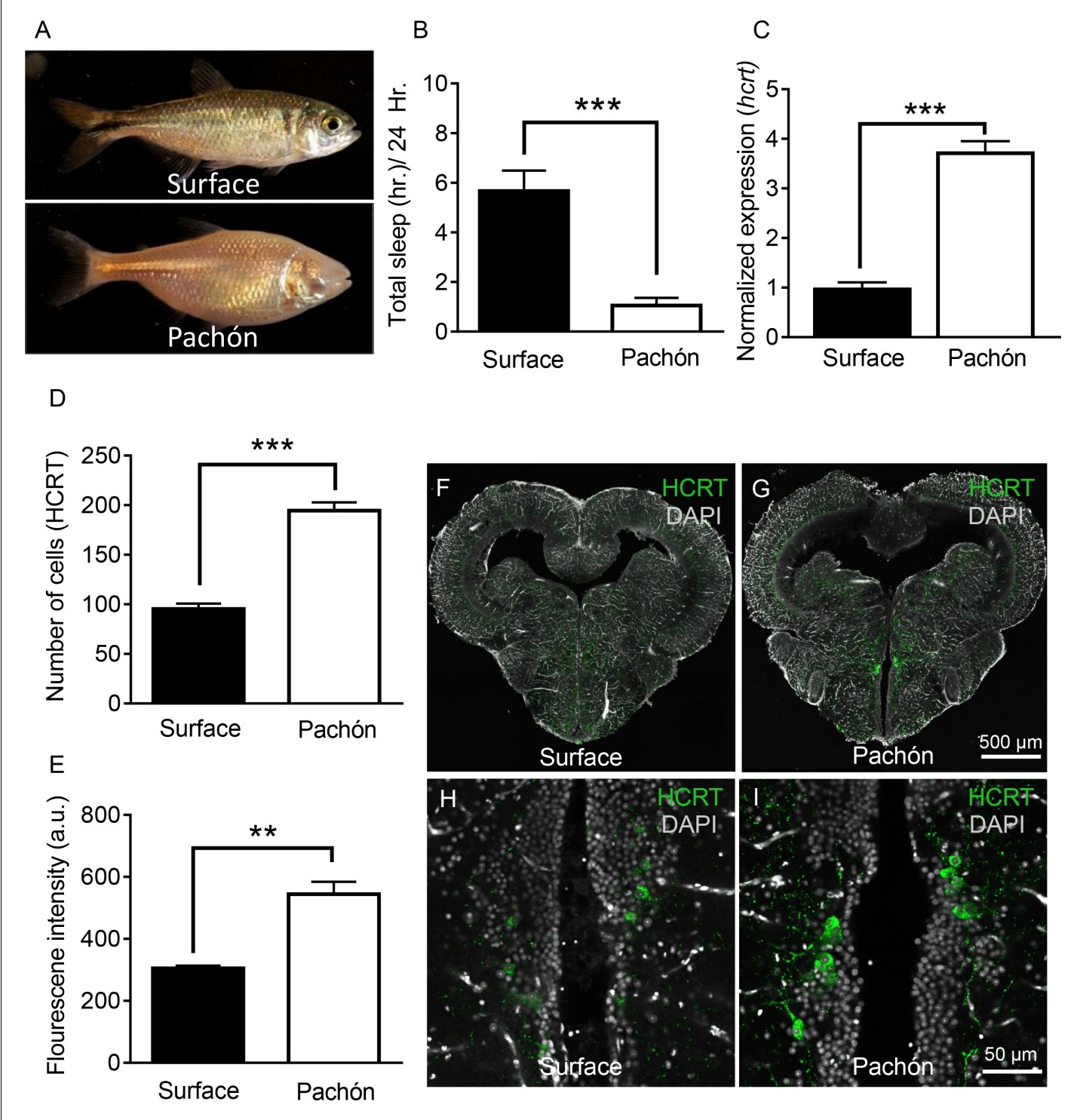

**Figure 1.** Hypocretin transcript and peptide levels are elevated in Pachón cavefish. (**A**) Representative images of surface fish and Pachón cavefish (**B**) Sleep duration is significantly reduced in Pachón cavefish compared to surface morph (Unpaired t-test, t = 5.56, n = 26, p<0.0001). (**C**) Expression of *hcrt* normalized by GAPDH in adult whole-brain extracts is significantly enhanced in Pachón cavefish compared to surface fish (Unpaired t-test, t = 11.15, n = 8, p<0.0001). (**D**) HCRT neuropeptide signal is significantly increased in Pachón cavefish compared to surface fish (Unpaired t-test, t = 5.94, n = 8, p<0.001). E. The number of HCRT-positive cells in the hypothalamus is significantly increased in cavefish compared to surface fish (Unpaired t-test, t = 9.984, n = 8, p<0.0001). (**F–I**) Representative 2 μm confocal images from coronal slices of surface fish or Pachón brains immunostained with anti-HCRT (green) and DAPI (white) (**F**) Surface whole brain coronal slice. (**G**) Pachón whole brain coronal slice. (**H**) Surface fish

*Figure 1 continued on next page*

*Figure 1 continued*

dorsal hypothalamus containing HCRT positive cells (I) Pachón cavefish dorsal hypothalamus containing HCRT neurons in view. Scale bar denotes 500 μm (F,G); 50 μm (H,I).

DOI: https://doi.org/10.7554/eLife.32637.002

The following source data and figure supplements are available for figure 1:

**Source data 1.** Hypocretin transcript and peptide levels are elevated in Pachón cavefish.

DOI: https://doi.org/10.7554/eLife.32637.006

**Figure supplement 1.** HCRT sequence is identical between surface fish and Pachón cavefish.

DOI: https://doi.org/10.7554/eLife.32637.003

**Figure supplement 1—source data 1.** Quantification of qPCR housekeeping genes and fluorescent intensity between surface and cavefish.

DOI: https://doi.org/10.7554/eLife.32637.007

**Figure supplement 2.** HCRT peptide levels are increased in Pachón cavefish.

DOI: https://doi.org/10.7554/eLife.32637.004

**Figure supplement 3.** Hypocretin levels are increased in early development in Pachón cavefish.

DOI: https://doi.org/10.7554/eLife.32637.005

suggesting the change in peptide levels were present at the time fish begin consuming food (*Figure 1—figure supplement 3*).

While mammals possess two HCRT receptors (HCRTR1 and HCRTR2), zebrafish only possess HCRTR2 (*Prober et al., 2006*; *Yokogawa et al., 2007*). Importantly, HCRTR2 is proposed to be evolutionarily more ancient compared to HCRTR1 (*Wong et al., 2011*). We performed genome analysis to explore how many paralog HCRTR genes resided in the genome of *A. mexicanus*, and found the cavefish and surface fish genomes encode only HCRTR2 (*McGaugh et al., 2014*).

To more directly assess the contributions of HCRT regulation in sleep loss, we measured the effect of HCRT receptor blockade on sleep in adult surface fish and Pachón cavefish. Fish from both populations were bathed in the selective HCRTR2 pharmacological inhibitor, TCSOX229 (*Kummangal et al., 2013*; *Plaza-Zabala et al., 2012*). Sleep in surface fish remained unchanged in the presence of 1 μM or 10 μM TCSOX229 (*Figure 2A,C*). Conversely, treatment of TCSOX229 in Pachón cavefish increased sleep duration compared to solvent treated (DMSO) controls (*Figure 2B, C*). While these results do not exclude the possibility that HCRT regulates sleep in surface fish, the sleep-promoting effect of TCSOX229 in Pachón cavefish suggests these fish are more sensitive to changes in HCRT signaling than surface fish. Treatment with TCSOX229 had no effect on waking velocity in surface fish or cavefish, suggesting that the increased quiescence observed in cavefish after drug treatment is not due to lethargy (*Figure 2D*). Further analysis revealed that sleep-promoting effects of TCSOX229 in cavefish can be attributed to both an increase in bout number and bout duration, suggesting that HCRT blockade affects sleep onset and maintenance (*Figure 2E,F*). Taken together, these findings support the notion that elevated HCRT signaling in cavefish underlies, in part, the evolution of sleep loss.

Sleep loss in *A. mexicanus* cavefish populations is found across developmental stages, from larval and juvenile forms to adults (*Duboué et al., 2011*; *Yoshizawa et al., 2015*). The small size of young fry (25 dpf) and ability to perform higher throughput analysis make them an excellent model for investigating the effects of drugs on sleep. Previous drug screens have been carried out using larval and juvenile zebrafish and *A. mexicanus* using standard concentrations between 1–30 μM for all drugs (*Duboué et al., 2011*; *Rihel et al., 2010*). We therefore selected additional pharmacological modulators of HCRTR2 based on permeability to the blood-brain barrier and affinity for HCRTR for testing in fry. Fish from both populations were bathed in the selective HCRTR2 pharmacological inhibitors, TCSOX229 (*Kummangal et al., 2013*; *Plaza-Zabala et al., 2012*), *N*-Ethyl-2-[(6-methoxy-3-pyridinyl) [(2-methylphenyl)sulfonyl]amino] -*N*-(3-pyridinylmethyl)- acetamide (EMPA) (*Malherbe et al., 2009*; *Mochizuki et al., 2011*), or the HCRTR1/2 antagonist Suvorexant (*Betschart et al., 2013*; *Hoyer et al., 2013*) (*Figure 3A*). A dose-response assay was carried out in juvenile fish for TCSOX229 (*Figure 3—figure supplement 2*) and found a significant effect in Pachón cavefish at a concentrations ranging from 1 to 30 μM. Therefore, all three antagonists were tested for their effect on sleep in surface and cavefish at a dose of 30 μM. None of the three antagonists altered sleep in surface fish, whereas they significantly increased sleep in cavefish (*Figure 3B*) at this concentration. The waking activity was not affected by antagonist treatment in surface fish or

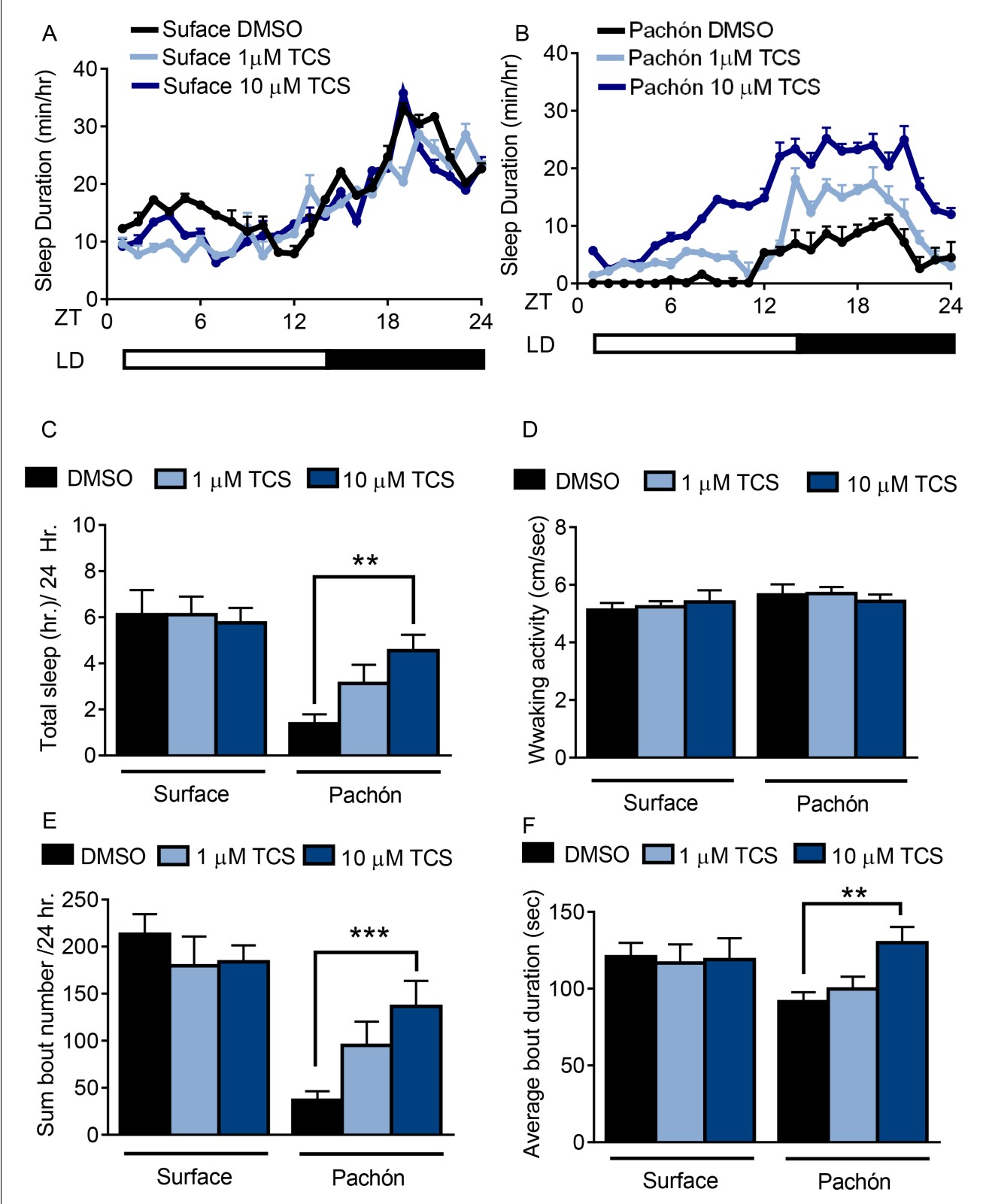

**Figure 2.** Pharmacological inhibition of HCRT Receptor two promotes sleep in Pachón cavefish. (A,B) Twenty-four hour sleep profile in surface fish (A) Pachón cavefish (B) treated with DMSO (black), 1 µm TCS (light blue) or 10 µm TCS (dark blue). (C) TCS treatment does not affect total sleep duration in surface fish (1 µM p>0.999, n = 12, 10 uM P>0.941, n = 12). Pachón cavefish teated with 1 µM TCS trended towards increased sleep (p>0.178, n = 12) while treatment with 10 µM significantly increased sleep (p<0.01, n = 13; F(1, 73)=25.00,) compared to control treated fish. (D) Waking activity was not

*Figure 2 continued on next page*

*Figure 2 continued*

significantly altered in surface fish or cavefish or in response to drug treatment, 2-way ANOVA, (F(1, 73)=2.73, p>0.103, n = 79) (**E**) Treatment with TCS did not affect average sleep bout duration in surface fish (1 µM TCS, p>0.430, n = 12; 10 µM TCS, p>0.518, n = 12) Treatment of Pachón cavefish with 1 µM TCS trended towards increased bout duration, p>0.051, n = 12, while 10 µM TCS treatment significantly increased bout duration in Pachón cavefish, (p<0.01, n = 13; F(1, 73)=47.42). (**F**) TCS treatment did not affect total sleep bout number in surface fish 1 µM TCS, p>0.976, n = 12; 10 µM TCS, p>0.998, n = 12). In Pachón cavefish, treatment with 1 µM TCS did not affect sleep bout number (p>0.828, n = 12). Treatment with 10 µM TCS significantly increased bout duration in Pachón cavefish,(p<0.001, n = 13; 2-way ANOVA, F(1, 68)=3.309).

DOI: https://doi.org/10.7554/eLife.32637.008

The following source data is available for figure 2:

**Source data 1.** Pharmacological inhibition of HCRT Receptor two promotes sleep in Pachón cavefish.

DOI: https://doi.org/10.7554/eLife.32637.009

cavefish, supporting the notion that the sleep promoting effects on cavefish are not due to induction of lethargy (*Figure 3C*). Further, all three HCRT antagonists significantly increased sleep bout number, and bout duration in cavefish (*Figure 3D,E*). While these results do not exclude the possibility that HCRT regulates sleep in surface fish, the sleep-promoting effect of HCRT antagonists in Pachón cavefish suggests these fish are more sensitive to changes in HCRT signaling than surface fish.

To determine the role of enhanced HCRTR2 signaling, we bathed 25 dpf fry in the HCRTR2 selective agonist YNT-185 (*Figure 3—figure supplement 1*). While the effect of this drug on sleep has not been previously tested in fish, it actively consolidates wakefulness in mice (*Irukayama-Tomobe et al., 2017*; *Nagahara et al., 2015*). Treatment with YNT-185 significantly reduced sleep in surface fish, without affecting sleep in cavefish where HCRT levels are naturally elevated (*Figure 3—figure supplement 1* ). Activity during waking bouts was not significantly altered with treatment of YNT-185 in either surface or Pachón cavefish, suggesting that the reduction in total sleep of surface fish was not due to lethargy. Further, YNT-185 treatment significantly reduced the total number of sleep bouts (*Figure 3—figure supplement 1*). Taken together, these findings support the notion that elevated HCRT signaling in larval and adult cavefish underlies the evolution of sleep loss.

To validate the sleep phenotypes obtained with pharmacological manipulation of HCRT signaling, we selectively knocked-down *hcrt* and measured the effect on sleep. Morpholinos (MOs) have been effectively used in zebrafish and *A. mexicanus* to knock-down gene function (*Bilandžija et al., 2013*; *Bill et al., 2009*). The knock-down effect of MO injection is typically limited to ~five days post injection, and we first verified that sleep differences are present at this early stage. We found that at four dpf, sleep in Pachón cavefish is significantly reduced compared to surface fish that were age-matched (*Figure 4A*). Injection of 0.2 mM *hcrt* MO enhanced sleep in cavefish compared to fish injected with scrambled MO control, whereas knock-down of *hcrt* using the same MO had no observable effect on surface fish (*Figure 4A*). The mortality of fish injected with 0.2 mM *hcrt* MOs, 0.2 mM scramble MOs, and non-injected controls did not differ, indicating at the concentration used, there is no generalized effect of injection procedure or MO treatment on survival (*Figure 4— figure supplement 1A–B*). While this baseline mortality is higher than zebrafish, it does not differ from standard *A. mexicanus* mortality observed in our lab or others, and therefore there does not appear to be a detrimental effect of injection procedure or treatment (*Elipot et al., 2014*). Morpholino treatment did not affect activity during wake bouts in surface fish or Pachón cavefish (*Figure 4B*). Analysis of sleep architecture revealed that injection of 0.2 mM *hcrt* MO increased total sleep bout number and sleep bout duration in Pachón cavefish, though not to levels of surface fish (*Figure 4C*; *Figure 4D*). Therefore, these findings support the notion that elevated levels of HCRT promote sleep in Pachón cavefish.

To validate further a role for HCRT in sleep regulation we sought to genetically silence HCRT neurons and assess sleep. The GAL4/UAS system has been widely in *Drosophila* and zebrafish to manipulate gene expression with spatial specificity (*Asakawa and Kawakami, 2008*; *Brand and Perrimon, 1993*; *Scheer and Campos-Ortega, 1999*). In zebrafish, co-injection of separate GAL4 and UAS plasmids flanked with the transposable element, Tol2 (*Kawakami et al., 2000*), has been effectively used to generate transient or stable expression in cells labeled by both transgenes (*Scott et al., 2007*). To silence HCRT neurons, we co-injected embryos with *hcrt*-GAL4 that drives expression in all HCRT neurons and UAS-*Botulinum toxin* (BoTX) that blocks neurotransmission by cleaving SNARE proteins required for synaptic release (*Brunger et al., 2008*). Embryos were injected, raised under

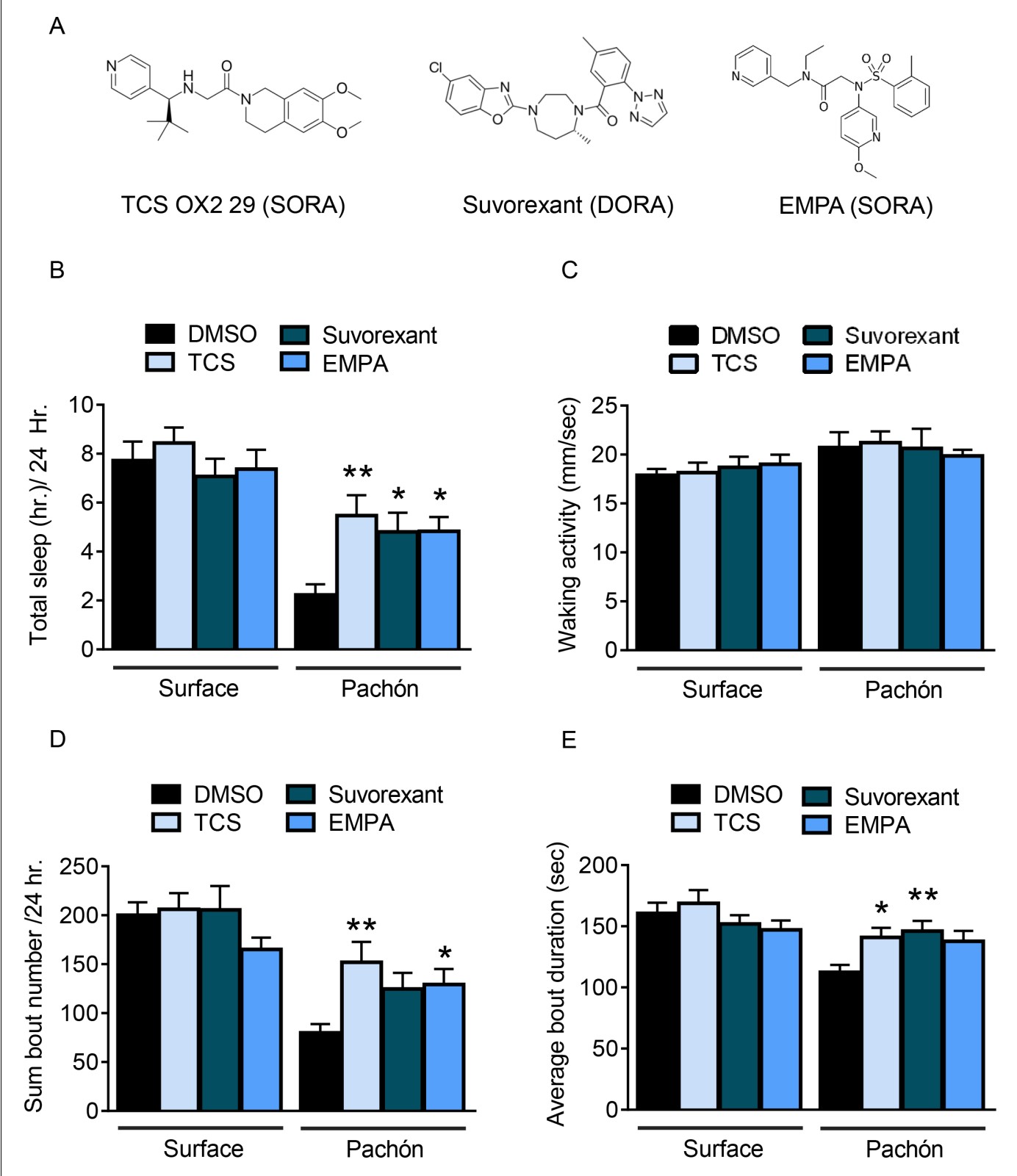

**Figure 3.** Panel of pharmacological HCRT antagonists reveals wake-promoting role of HCRT in juvenile cavefish. (**A**) Two SORAs and one DORA were used to inhibit binding of HCRT to its receptor. (**B**) Total sleep is not significantly altered in surface fish with the treatment of HCRT antagonists (TCS, p=0.814, n = 18; Suvorexant, p=0.865, n = 18; EMPA, p=0.972, n = 19). Pachón cavefish significantly increased total sleep in response to HCRT antagonists (TCS, p=0.004, n = 18; Suvorexant, p=0.032, n = 19, EMPA, p=0.027, n = 18; 2-way ANOVA, F(1, 135)=43.70, p<0.0001). (**B**) Waking activity

*Figure 3 continued on next page*

*Figure 3 continued*

was not significantly altered in any fish or in response to drug treatment, Surface TCS, p=0.996, n = 18; Suvorexant, p=0.925, n = 18; EMPA, p=0.816, n = 19; Pachón, TCS, p=0.980, n = 18; Suvorexant, p>0.999, n = 19; EMPA, p=0.917, n = 18; 2-way ANOVA, (F$_{(1, 135)}$=7.21, p=0.008) (C) The total number of sleep bouts over 24 hr of drug treatment did not significantly change in surface fish (TCS, p=0.814, n = 18; Suvorexant, p=0.865, n = 18; EMPA, p=0.972, n = 19). Total sleep bouts were significantly increased in Pachón cavefish with TCS, p=0.002, n = 18; and EMPA, p=0.041, n = 18, Suvorexant trended towards significance, p=0.071, n = 19 (2-way ANOVA, F(1, 135)=45.27, p<0.0001). (D) Average sleep bout length was not significantly different in Surface fish treated with HCRT antagonists (TCS, p=0.822, n = 18; Suvorexant, p=0.805, n = 18; EMPA, p=0.236, n = 19). Pachón cavefish significantly increased their average sleep bout lengths with TCS, p=0.039, N = 18; and with Suvorexant, p=0.009, n = 19. EMPA trended towards significance, p=0.078, n = 18 (2-way ANOVA, F(1, 135)=14.06, p=0.0003).
DOI: https://doi.org/10.7554/eLife.32637.010

The following source data and figure supplements are available for figure 3:

**Source data 1.** Panel of pharmacological HCRT antagonists reveals wake-promoting role of HCRT in juvenile cavefish.
DOI: https://doi.org/10.7554/eLife.32637.013
**Figure supplement 1.** HCRT agonist partially alters sleep behavior in juvenile surface fish, but not Pachón cavefish.
DOI: https://doi.org/10.7554/eLife.32637.011
**Figure supplement 1—source data 1.** Inhibition of HCRT signaling increases sleep in larval cavefish.
DOI: https://doi.org/10.7554/eLife.32637.014
**Figure supplement 2.** HCRTR2 blockade selectively increases sleep in Pachón cavefish.
DOI: https://doi.org/10.7554/eLife.32637.012
**Figure supplement 2—source data 2.** HCRT agonist partially alters sleep behavior in juvenile surface fish, but not Pachón cavefish.
DOI: https://doi.org/10.7554/eLife.32637.015

standard conditions, then tested for sleep at 25 dpf. Following sleep measurements, brains of individual fish were dissected and the number of silenced HCRT neurons (identifiable by GFP expression) were quantified. In all cases, no expression was observed in fish injected with UAS-*BoTXBLC-GFP* alone, indicating that *hcrt*-GAL4 is required for driving expression of the transgene (*Figure 5A, C*). Further, all GFP-positive neurons were co-labeled by Anti-HCRT, demonstrating that this approach specifically targets *BoTXBLC* transgene expression to HCRT neurons (*Figure 5B,D*). The total sleep in fish expressing *hcrt*:GAL4; UAS-*BoTxTxBLC-GFP* were compared to wild type fish or fish injected with UAS-*BoTxTxBLC-GFP* alone. Sleep was significantly increased in experimental double transgenic Pachón cavefish (*hcrt*:GAL4:UAS-*BoTxTxBLC-GFP*) compared to both control groups, while there was no effect in surface fish (*Figure 5E*). Waking velocity in surface fish and cavefish was not changed between both control groups and fish expressing *hcrt*:GAL4; UAS-*BoTxBLC-GFP*, suggesting that the increased quiescence from neuronal silencing is not due to lethargy (*Figure 5F*).

Silencing subsets of HCRT neurons increased sleep in Pachón cavefish by increasing bout number (*Figure 5A–S*) without affecting bout duration (*Figure 5B–S*). Quantification of labeled cells revealed 10.2% of hcrt:GAL4; UAS-BoTxTxBLC-GFP cavefish express GFP in an average of 7.5 cells per expressing animal. Similarly, 11.5% of hcrt:GAL4; UAS-BoTxTxBLC-GFP surface fish express GFP with an average of 6.7 cells per expressing animal. Regression analysis of the number of neurons silenced significantly correlated with total sleep duration (R$^2$ = 0.229) in Pachón cavefish, fortifying the notion that increased HCRT signaling is associated with sleep loss (*Figure 5G*). A weak correlation between BoTX-expressing HCRT neurons and sleep duration (R$^2$ = 0.091) was also observed in surface fish *Figure 5G*, consistent with the notion that HCRT may also regulates sleep in surface fish, but cavefish are more sensitive to subtle changes of HCRT signaling.

Hypocretin neurons are modulated by sensory stimuli and feeding state, indicating that they are involved in the integration of environmental cues with sleep regulation (*Appelbaum et al., 2007*; *Mileykovskiy et al., 2005*). The number of mechanosensory neuromasts that comprise the lateral line, a neuromodulatory system used to detect prey and water flow, are increased in cavefish. This evolved trait is hypothesized to allow for an enhanced ability to forage, object detection, and social behaviors in the absence of eyes (*Kowalko et al., 2013*; *Kulpa et al., 2015*; *Yoshizawa et al., 2010*). We previously reported that ablation of the lateral line restores sleep to Pachón cavefish without affecting sleep in surface fish, raising the possibility that lateral line input modulates HCRT signaling in cavefish to suppress sleep (*Jaggard et al., 2017*). To investigate the effects of lateral line input on HCRT, we pre-treated adult fish in the ototoxic antibiotic gentamicin, which effectively ablates the lateral line (*Van Trump et al., 2010*), and assayed sleep in adult cave and surface fish. In

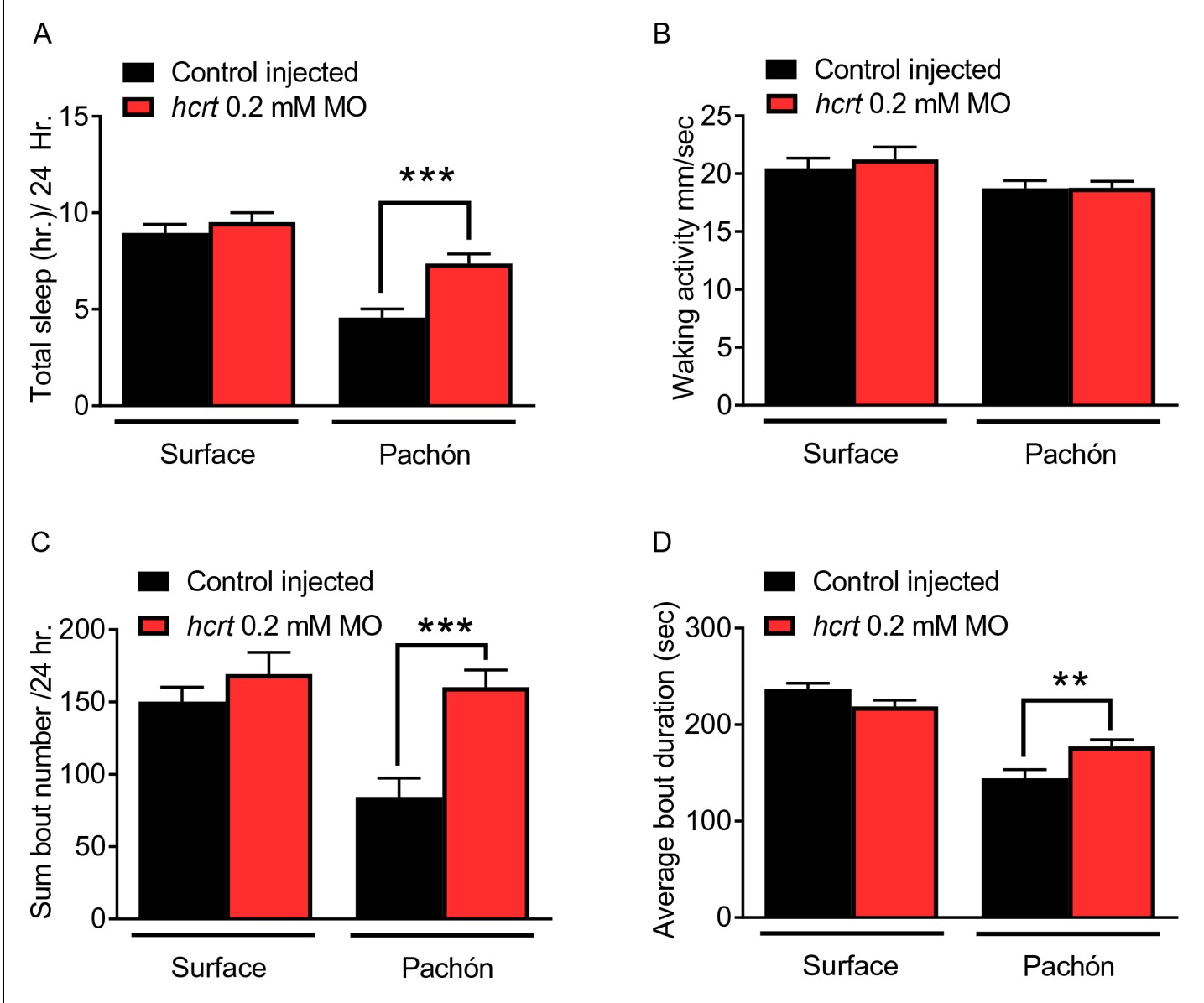

**Figure 4.** Transient knockdown of *hcrt* increases sleep in four dpf cavefish. (**A**) Morpholino knock-down of *hcrt* does not alter total sleep in surface fish, p=0.640, n = 34, while in cavefish total sleep time was significantly increased with *hcrt* knock-down, p=0.0002, n = 40; 2-way ANOVA, F(1,133)=45.82, p<0.0001 (**B**) Waking activity was not significantly altered in either surface fish (p=0.343, n = 34) or Pachón cavefish (p=0.084, n = 40; 2-way ANOVA, F(1, 133)=5.807). (**C**) Total number of sleep bouts in surface fish was not significantly different from injected controls, p=0.459, n = 34. While in Pachón cavefish, total sleep bouts over 24 hr was significantly increased in *hcrt* MO fish compared to control fish (p=0.0004, n = 40; 2-way ANOVA, F(1,133) =8.295, p=0.004). (**D**) Average sleep bout duration was not different between controls and *hcrt* MO injected surface fish (p=0.081, n = 34). There was a significant increase in average bout duration In Pachón *hcrt* MO injected fish compared to their respective controls (p=0.004, n = 40; 2-way ANOVA, F (1,133)=13.61, p=0.0003).

DOI: https://doi.org/10.7554/eLife.32637.016

The following source data and figure supplements are available for figure 4:

**Source data 1.** Transient knockdown of *hcrt* increases sleep in 4 days post fertilization cavefish.

DOI: https://doi.org/10.7554/eLife.32637.017

**Figure supplement 1.** Morpholino injections do not affect survival in development.

DOI: https://doi.org/10.7554/eLife.32637.018

**Figure supplement 1—source data 1.** The effect of morpholino injection on survival in surface fish and cavefish.

DOI: https://doi.org/10.7554/eLife.32637.019

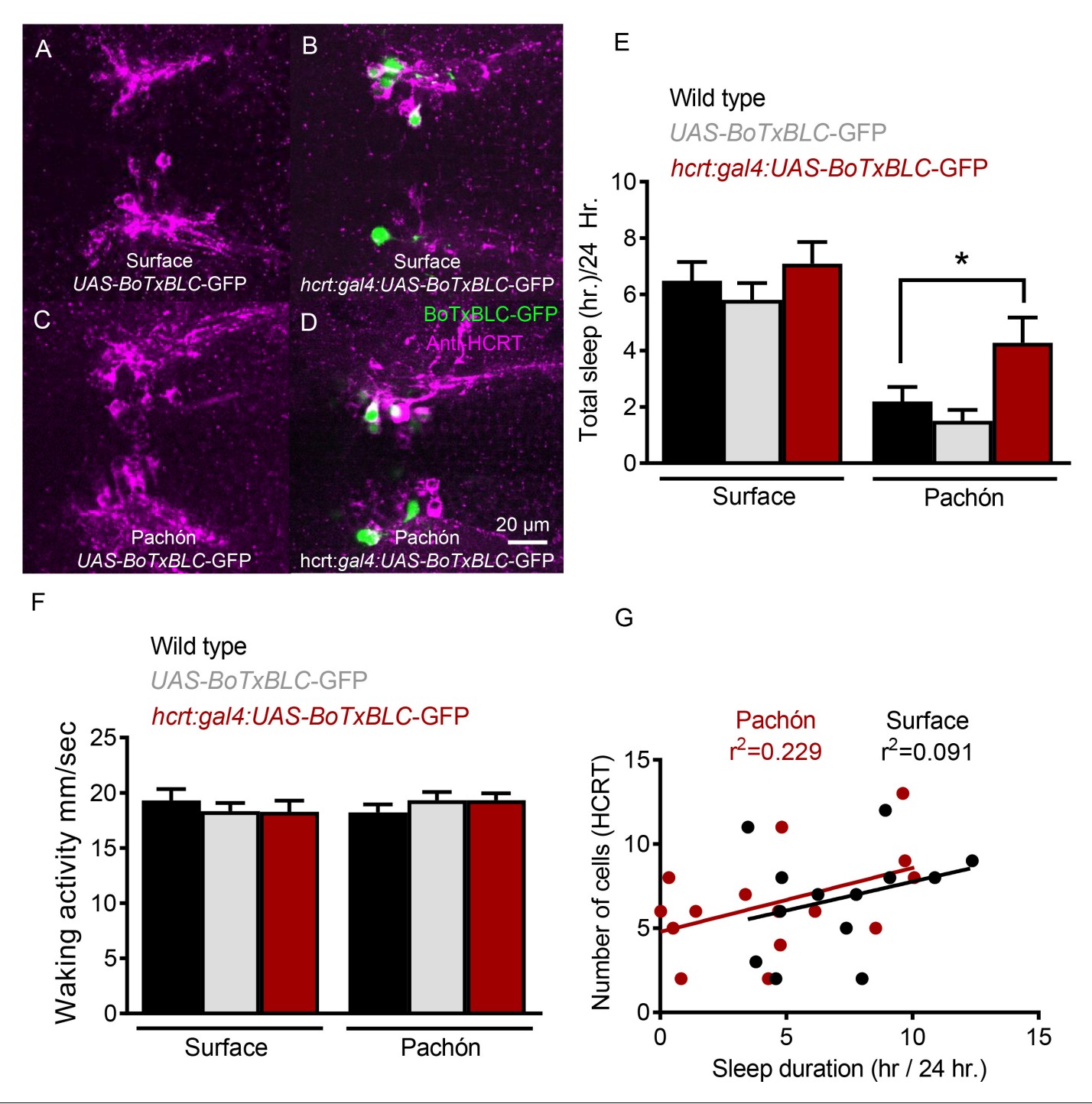

**Figure 5.** Genetic silencing of subsets of HCRT neurons selectively increases sleep in Pachón cavefish. (**A**) Surface UAS-*BoTx-BLC-GFP* (**B**) Surface *hcrt*:GAL4-UAS-*BoTx-BLC-GFP* (**C**) Pachón *UAS-BoTx-BLC-GFP* (**D**) Pachón *hcrt*:GAL4-UAS-*BoTx-BLC-GFP* (**E**) Neuronal silencing of HCRT cells did not alter total sleep duration in surface fish (Wild Type*UAS-*BoTx-BLC-GFP*, p=0.989, n = 23; UAS-*BoTx-BLC-GFP** *hcrt*:GAL4-UAS-*BoTx-BLC-GFP*, p=0.713, n = 14). Pachón cavefish increased their total sleep when HCRT cells were silenced (Wild Type*UAS-*BoTx-BLC-GFP*, p=0.667, n = 21; UAS-*BoTx-BLC-GFP** *hcrt*:GAL4-UAS-*BoTx-BLC-GFP*, p=0.0426, n = 15; 2-way ANOVA, F(1,91)=65.68, p<0.0001). (**F**) Waking activity was not altered in either Surface or Pachón cavefish with neuronal silencing of HCRT (Surface fish Wild Type*UAS-*BoTx-BLC-GFP*, p=0.616, n = 23; UAS-*BoTx-BLC-GFP** *hcrt*:GAL4-UAS-*BoTx-BLC-GFP*, p=0.642, n = 14; Pachón Wild Type*UAS-*BoTx-BLC-GFP*, p=0.587 n = 21; UAS-*BoTx-BLC-GFP** *hcrt*:GAL4-UAS-*BoTx-BLC-GFP*, p=0.612, n = 15; 2-way ANOVA, F(1,91)=0.206, p=0.650). (**G**) Regression analysis revealed there was a subtle trend of increased sleep with more HCRT neurons silenced as quantified with GFP signal (R2 = 0.091, p=0.314, n = 14). The same regression analysis in Pachón cavefish revealed a much more robust correlation to increased sleep in more silenced HCRT cells (R2 = 0.229, p=0.0871, N = 15).

*Figure 5 continued on next page*

*Figure 5 continued*

DOI: https://doi.org/10.7554/eLife.32637.020

The following source data and figure supplements are available for figure 5:

**Source data 1.** Genetic silencing of subsets of HCRT neurons selectively increases sleep in Pachón cavefish.
DOI: https://doi.org/10.7554/eLife.32637.022
**Figure supplement 1.** Genetic silencing of subsets of HCRT alters sleep architecture in Pachón cavefish.
DOI: https://doi.org/10.7554/eLife.32637.021
**Figure supplement 1—source data 1.** Genetic silencing of subsets of HCRT alters sleep architecture in Pachón cavefish.
DOI: https://doi.org/10.7554/eLife.32637.023

agreement with previous findings, gentamicin treatment fully ablated the lateral line (*Figure 6A–D*) and restored sleep in cavefish without affecting sleep in surface fish (*Jaggard et al., 2017*). To determine the effect of lateral line ablation on HCRT regulation, we quantified *hcrt* expression or neuropeptide levels in adult cave and surface fish following gentamicin treatment. Quantitative PCR analysis of gentamicin treatment revealed that *hcrt* expression was significantly reduced in cavefish to levels equivalent to untreated surface fish (*Figure 6E*). By contrast, there were no significant changes in *hcrt* expression following gentamicin treatment in surface fish. Therefore, the robust effect of lateral line ablation in cavefish indicates that the lateral line selectively enhances HCRT levels in cavefish. Comparison and detailed quantification of HCRT neuropeptide levels in the hypothalamus reveals that lateral line ablation does not impact the number of HCRT-positive hypothalamic neurons, but instead selectively reduces the level of HCRT within each cell in cavefish (*Figure 6F–K*), supporting the notion that the lateral line is required for enhancement of HCRT function in cavefish. Together, these findings reveal that sensory input from the lateral line promotes sleep and *hcrt* expression in Pachón cavefish, providing a link between sensory input and transcriptional regulation of a wake-promoting factor.

In addition to its potent role in sleep regulation, *hcrt* promotes food consumption in fish and mammals (*Penney and Volkoff, 2014*; *Tsujino and Sakurai, 2013*; *Yokobori et al., 2011*). We previously reported that prolonged starvation increases sleep in cavefish without affecting sleep in surface fish (*Jaggard et al., 2017*), but the role of HCRT in feeding-state dependent modulation of sleep-wake cycles has not been investigated. Quantitative PCR analysis from whole-brain extracts revealed that *hcrt* transcript is significantly reduced in cavefish following 30 days of starvation; however, the same treatment does not affect *hcrt* transcription in surface fish, indicating that cavefish are more sensitive to starvation-dependent changes in HCRT (*Figure 7A*). To determine whether HCRT neuropeptide is produced in a greater number of cells during starvation, we quantified HCRT-positive neurons in fed and starved state (*Figure 7B–G*). Similar to lateral line ablation, starvation reduced HCRT levels in each cell, without affecting the number of HCRT-positive neurons. Further, starvation did not affect the number of HCRT-positive cells or HCRT levels per cell in surface fish (*Figure 7B*). These results indicate that the starvation modulates HCRT levels, rather than the number of cells that produce HCRT. The acute regulation of HCRT by feeding state and lateral line dependent sensory input demonstrates a unique link between these neuronal systems and those mediating sleep/wake cycles.

## Discussion

Cavefish are a unique model for investigating neural and genetic regulation of sleep, particularly from an evolutionary perspective. Robust phenotypic differences have been observed in multiple populations of cavefish, but our findings provide the first evidence of altered regulation of a neuromodulatory peptide that associates with the evolution of sleep loss. Alignment of *hcrt* sequences derived from surface and Pachón cavefish indicate that there are no differences in the genomic sequences within coding regions of the two morphs. Our findings do, however, reveal dramatic differences in *hcrt* expression and neuron number between surface fish and cavefish, raising the possibility that regulation and development of *hcrt* is altered evolutionarily. These observations are in agreement with concurrent findings that increased levels of the homeobox transcription factor Lhx9 specifies greater number of HCRT-neurons in cavefish (*Alie et al., 2018*). Further, the acute differences in *hcrt* expression between surface fish and cavefish likely occur at the level of changes in

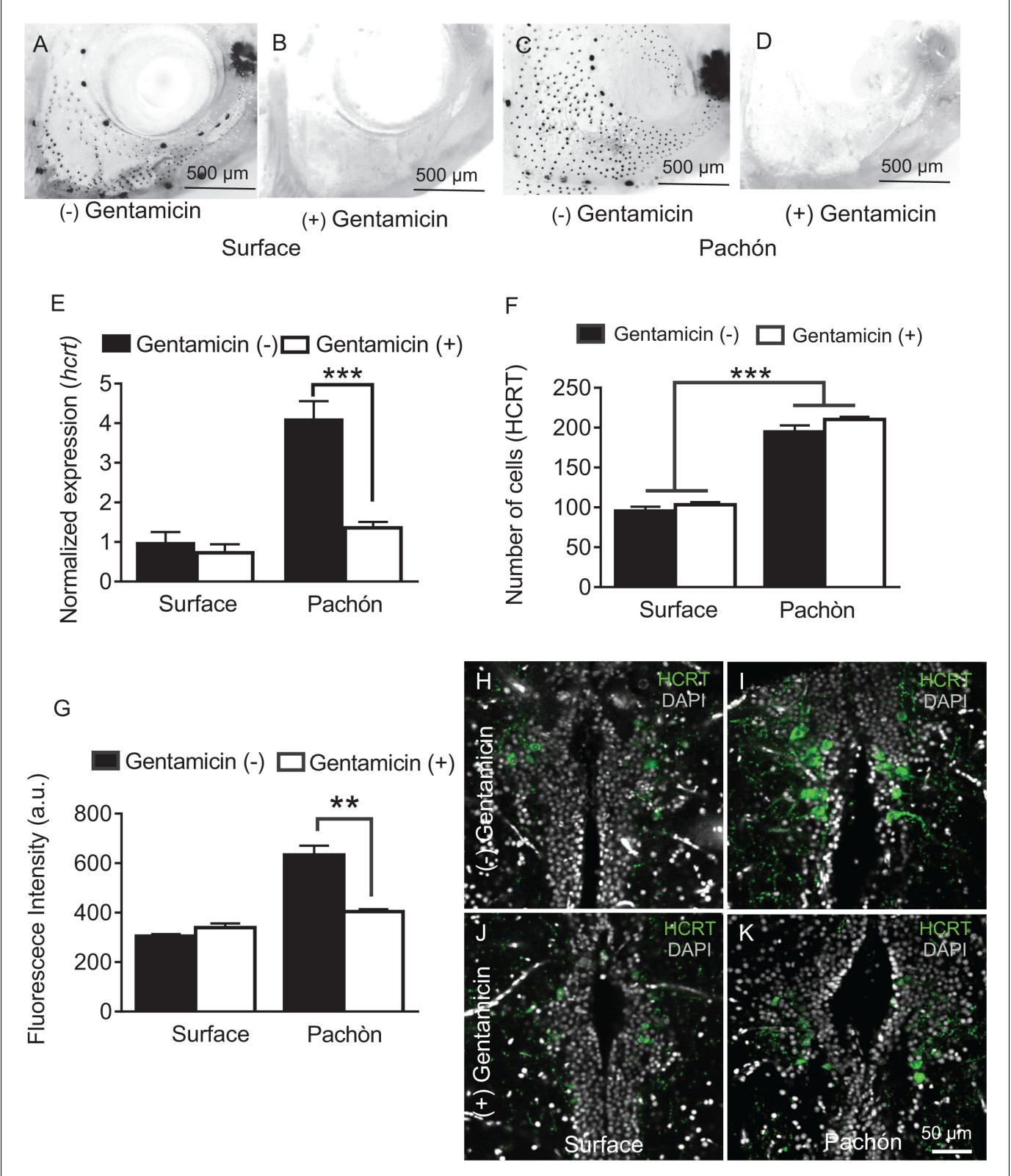

**Figure 6.** Chemical ablation of mechanosensory lateral line reduces HCRT levels in Pachón cavefish. (A-D). Photomicrographs of surface fish cranial regions stained with DASPEI to reveal lateral line mechanosensory neuromasts. Treatment with gentamicin ablates lateral line neuromasts in surface fish (B) and Pachón cavefish (D) (E) Gentamicin treatment has no significant effect on *hcrt* expression in surface fish (p>0.635, n = 8) while in Pachón cavefish gentamicin treatment significantly reduces *hcrt* expression, restoring surface-like levels. (Pachón treated vs. untreated, p<0.0001; Pachón treated vs.

*Figure 6 continued on next page*

*Figure 6 continued*

surface untreated, p>0.635, n = 8, F(1,28)=21.28). (F) Fluorescent intensity per hypothalamic HCRT-cell was not altered with gentamicin treatment in surface fish, p=0.590, n = 4. In Pachón cavefish, HCRT neuropeptide levels are significantly lower following gentamicin treatment (p<0.0001, n = 4; 2-way ANOVA, F (1, 13)=0.0001 G. Gentamicin treatment has no effect on total number of HCRT cell number in either surface or Pachón cavefish (p>0.494, n = 8, F (1, 13C)=0.4967). (H–K) Representative 2 µm confocal images of the dorsal hypothalamic region in surface fish and Pachón cavefish immunostained with HCRT (green) and DAPI (white) (H) Surface control (I) Pachón control (J) Surface gentamicin K. Pachón gentamicin. Scale bar = 50 µm.

DOI: https://doi.org/10.7554/eLife.32637.024

The following source data is available for figure 6:

**Source data 1.** Chemical ablation of mechanosensory lateral line reduces HCRT levels in Pachón cavefish.

DOI: https://doi.org/10.7554/eLife.32637.025

genomic enhancers or neuronal connectivity, which affect HCRT functioning. Because our findings also reveal an increased number of HCRT-positive neurons in early development, it is also likely that developmental differences between the brains of surface and cavefish underlie differences in HCRT function.

Examination of cell body number in 6 dpf fry reveals increased HCRT-positive neurons in cavefish, indicating HCRT differences are present during early development. In agreement with these findings, broad anatomical differences in forebrain structure have previously been documented between surface fish and cavefish including an expanded hypothalamus (*Menuet et al., 2007*). A concurrent paper revealed that neurons expressing Neuropeptide Y are also increased in cavefish (*Alie et al., 2018*). Like Hcrt, Neuropeptide Y also regulates feeding and sleep in diverse species, suggesting changes in hypothalamic regulation of sleep are, at least partially, developmentally-derived and not limited to the changes in Hcrt function that we have observed (*Chung et al., 2017*; *Luquet et al., 2005*; *Szentirmai and Krueger, 2006*; *Wu et al., 2003*). Therefore, it is likely that developmentally-derived differences in the number of HCRT-positive neurons and modified hypothalamic neural circuitry contribute to sleep loss in cavefish.

Multiple lines of evidence presented here support a robust role for evolved differences in HCRT function in the evolution of sleep loss in Pachón cavefish. Pharmacological inhibition of HCRT signaling with three different HCRTR2 antagonists restored sleep in cavefish, suggesting that HCRTsignaling is required for at least some of the sleep loss in cavefish. The antagonists used have not been validated in zebrafish or *A. mexicanus*, and therefore it is possible that non-specific effects contribute to the sleep phenotype. Future work characterizing these pharmacological inhibitors in *A. mexicanus*, will be useful for validating the phenotypes observed in this manuscript. Complementing pharmacological experiments, targeted knockdown of *hcrt* using morpholinos, or genetic silencing of HCRT neurons promote sleep in Pachón cavefish without significantly affecting sleep in surface fish. While manipulations that inhibit HCRT signaling have potent affects on cavefish sleep, it does not rule out a possible role for HCRT modulation of sleep in surface fish. All three manipulations employed are likely to only partially disrupt HCRTR signaling, and it is likely that complete inhibition of HCRTR signaling would increase sleep in surface fish. Indeed, Hypocretin is highly conserved and has been shown to consolidate wakefulness in species including in animals ranging from zebrafish to humans (*Appelbaum et al., 2009*; *Chemelli et al., 1999*; *Mileykovskiy et al., 2005*; *Prober et al., 2006*). Moreover, the Hypocretin system has been characterized in detail in zebrafish and acts as a key mediator of arousal (*Appelbaum et al., 2009*; *Elbaz et al., 2012*; *Kaslin et al., 2004*; *Prober et al., 2006*). The findings presented here extend studies in mammalian and zebrafish models, and suggest that regulation of HCRT signaling may be subject to evolutionary pressure, and implicate it as a potential 'hot-spot' for variation in sleep throughout the animal kingdom.

While the neural processes regulating HCRT activity are not fully understood, growing evidence suggests these neurons integrate sleep-wake regulation with responses to sensory stimuli (*Appelbaum et al., 2007*; *Mileykovskiy et al., 2005*; *Woods et al., 2014*). In mice, HCRT neurons are transiently activated by sound, feeding, and cage exploration, suggesting that HCRT neurons are generally regulated by external stimuli (*Mileykovskiy et al., 2005*). Further in zebrafish, HCRT neurons activation is associated with periods of wakefulness and overexpression of HCRT enhances locomotor response to diverse sensory stimuli, while ablation of HCRT neurons reduces response to sound stimulus (*Elbaz et al., 2012*; *Naumann et al., 2010*; *Prober et al., 2006*; *Woods et al.,*

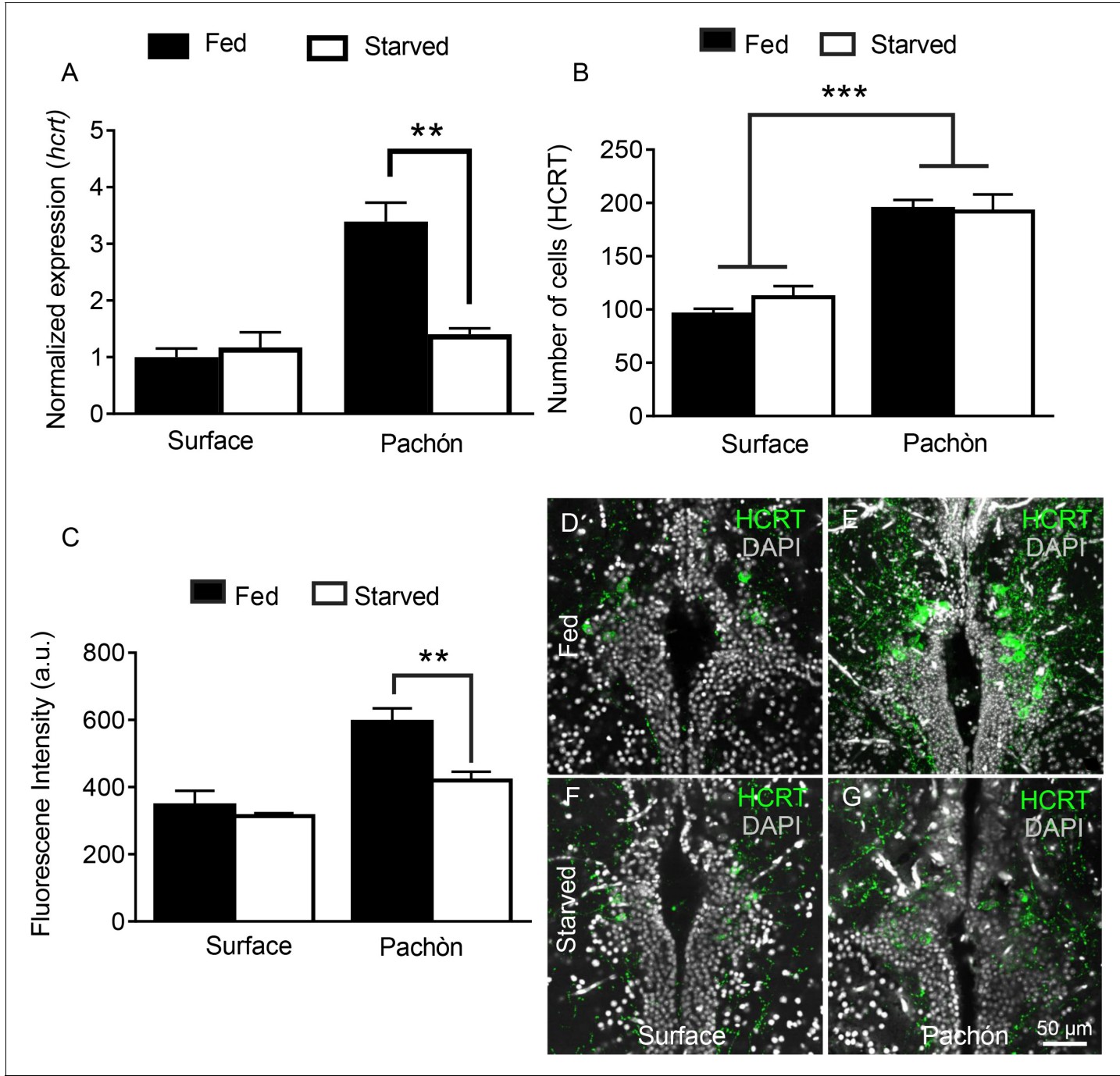

**Figure 7.** Starvation selectively inhibits HCRT levels in cavefish. (**A**) Starvation does not affect *hcrt* expression in surface fish (p>0.832, n = 4) while *hcrt* expression is significantly reduced in Pachón cavefish (p >0.001, n = 4, 2-way ANOVA, F(1,13)=13.54)) (**B**) Fluorescent intensity in HCRT cells was not affected by 30 days starvation in surface fish (p>0.788, n = 4). In Pachón cavefish, HCRT neuropeptide was significantly reduced following starvation (p<0.004, n = 4, 2-way ANOVA, F(1,12)=10.17)) (**C**) Starvation has no significant effect on total number of HCRT-positive cells in either surface or Pachón cavefish (Surface, p=0.452, n = 4; Pachón, p>0.979, n = 4, 2-way ANOVA, F(1,11)=3.65)) (**D**) Surface control (**E**) Pachón control (**F**) Surface starved (**G**) Pachón Starved. Scale bar = 50 μm.

DOI: https://doi.org/10.7554/eLife.32637.026

The following source data is available for figure 7:

**Source data 1.** Starvation selectively inhibits HCRT levels in cavefish.

DOI: https://doi.org/10.7554/eLife.32637.027

*2014*), suggesting that HCRT neurons mediate sensory responsiveness and sleep-wake behavior. Our findings reveal that ablation of the lateral line in cavefish reduces *hcrt* transcript and HCRT neuropeptide abundance to levels indistinguishable from their surface fish conspecifics, indicating that lateral line input is a potent regulator of *hcrt* production in cavefish. These findings support the notion that evolution of sensory systems dramatically affect central brain processes that regulate behavior, and provide further support that HCRT neurons integrate sensory stimuli to modulate sleep and arousal.

While a full understanding of the neural circuitry regulating HCRT-positive neurons has not been determined, HCRT neurons send projections to numerous areas implicated in behavioral regulation including the periventricular hypothalamus, the raphe, and thalamic nuclei (*Panula, 2010*). Evidence suggests that the wake-consolidating role of HCRT neurons is dependent on norepinephrine signaling, and optogenetic activation of HCRT neurons activates the locus coeruleus (*Singh et al., 2015*), raising the possibility that activation of this arousal pathway is enhanced in Pachón cavefish. Norepinephrine levels are elevated in the brains of cavefish and treatment of cavefish with the β-adrenergic inhibitor propranolol restores sleep in cavefish without affecting sleep in surface fish. (*Bilandžija et al., 2013*; *Duboué et al., 2012*). Therefore, it is possible that differences in norepinephrine signaling contribute to *hcrt*-dependent sleep loss in cavefish. Further investigation of the synergistic effects of norepinephrine and *hcrt*, and the effects of their pre-supposed interaction on feeding- and sensory-mediated *hcrt* regulation, will be critical in our understanding of how sleep changes can be driven by alterations in the environment.

The finding that ablation of the lateral line or starvation suppress *hcrt* transcript reveals a role for plasticity of HCRT signaling in response to environmental perturbation. In addition to its role in sleep-wake regulation, *hcrt* neurons regulate feeding and metabolic function, raising the possibility that HCRT neurons are integrators of sleep and metabolic state. Previous findings reveal that injection of HCRT peptide increases food consumption in cavefish, suggesting the consummatory behavior induced by HCRT in mammals is conserved in *A. mexicanus* (*Penney and Volkoff, 2014*; *Wall and Volkoff, 2013*). Studies in mammals and zebrafish suggest HCRT neurons are regulated by the adipose peptide hormone, Leptin (*Leinninger et al., 2011*; *Levitas-Djerbi et al., 2015*). Adipose levels in cavefish are elevated compared to surface fish (*Aspiras et al., 2015*), and it is possible that prolonged starvation reduces Leptin levels, thereby inhibiting *hcrt* expression to promote sleep in cavefish. Further, the lateral line system is intimately linked to feeding behavior in fish, and evolved differences in the lateral line system have been identified between surface fish and cavefish (*Yoshizawa et al., 2012*), perhaps including increased modulation of *hcrt* neurons. These findings raise the possibility of using cavefish as a model for examining the Leptin-HCRT axis and, more generally, interactions between sleep, sensory processing and metabolic function.

Our findings specifically examine neural mechanisms underlying sleep loss in the Pachón cave populations. Both morphological and genomic data suggest Pachón cavefish are one of the oldest, and most troglomorphic of the 29 *A. mexicanus* cavefish populations (*Bradic et al., 2012*; *Dowling et al., 2002*; *Ornelas-García et al., 2008*; *Strecker et al., 2003*). We have also demonstrated evolutionary convergence on sleep loss in other populations of cavefish, including Molino, Tinaja and Chica cave populations (*Jaggard et al., 2017*). However, ablation of the lateral line has no effect on sleep in Molino, Tinaja and Chica populations, suggesting distinct neural mechanism underlie sleep loss between Pachón cavefish, and other cavefish populations assayed (*Jaggard et al., 2017*). Future studies will reveal if enhanced HCRT function represents a conserved mechanism for sleep loss, or that sleep loss in other fish populations is HCRT independent.

Taken together, these studies raise the possibility that evolved differences in HCRT function contribute to the evolution of sleep loss in cavefish. Sleep is highly pleiotropic and it is likely that many additional genetic factors contribute to sleep differences between sleep in cavefish. Zebrafish provide a comparable genetic model that has implicated many novel genes and modulators in sleep regulation including Histamine, Corticoptropin Releasing Hormone and melatonin (*Appelbaum et al., 2009*; *Chiu et al., 2016*; *Kaslin et al., 2004*; *Zhdanova et al., 2001*). Future work examining functional differences in these genes may provide a mechanism for identifying additional genes involved in the evolution of sleep. In addition, the interfertility between surface and cave populations of *A. mexicanus* allows for QTL mapping and the potential to identify novel genetic regulators of sleep (*Gross, 2012*; *Jeffery, 2009*). The identification of evolved differences in HCRT

that likely regulate sleep provide a starting point for elucidating the genetic underpinnings associated with the evolution of complex behaviors.

## Materials and methods

### Fish maintenance and rearing

Animal husbandry was carried out as previously described (*Borowsky, 2008b*) and all protocols were approved by the IACUC Florida Atlantic University (Protocols A15-32 and A16-04). Fish were housed in the Florida Atlantic University core facilities at 23 ± 1°C constant water temperature throughout rearing for behavior experiments (*Borowsky, 2008b*). Lights were kept on a 14:10 hr light-dark cycle that remained constant throughout the animal's lifetime. Light intensity was kept between 25–40 Lux for both rearing and behavior experiments. All fish used for experiments were raised to adulthood and housed in standard 18–37L tanks. Adult fish were fed a mixture diet of black worms to satiation twice daily at zeitgeber time (ZT) 2 and ZT12, (Aquatic Foods, Fresno, CA,) and standard flake fish food during periods when fish were not being used for behavior experiments or breeding (Tetramine Pro).

### Sleep behavior

Adult fish were recorded in standard conditions in 10L tanks with custom-designed partitions that allowed for five fish (2L/fish) to be individually housed in each tank as previously described (*Yoshizawa et al., 2015*). Recording chambers were illuminated with custom-designed IR LED source (Infrared 850 nm 5050 LED Strip Light, Environmental Lights). After a 4–5 day acclimation period, behavior was recorded for 24 hr beginning ZT0-ZT2. Videos were recorded at 15 frames/sec using a USB webcam (LifeCam Studio 1080 p HD Webcam, Microsoft) fitted with a zoom lens (Zoom 7000, Navitar). An IR high-pass filter (Edmund Optics Worldwide) was placed between the camera and the lens to block visible light. For larval fish recordings, individual fish were placed in 12 well tissue culture plates (BD Biosciences). Recording chambers were lit with a custom-designed IR LED light strip and placed beneath the recording platform. Fish were allowed to acclimate for 24 hr before starting behavioral recordings. Videos were recorded using Virtualdub, a video-capturing software (Version 1.10.4) and were subsequently processed using Ethovision XT 9.0 (Noldus, IT). Water temperature and chemistry were monitored throughout recordings, and maintained at standard conditions in all cases. Ethovision tracking was setup as previously described (*Yoshizawa et al., 2015*). Data was then processed using Perl scripts (v5.22.0, developed on-site) and Excel macro (Microsoft) (*Yoshizawa et al., 2015*). These data were used to calculate sleep information by finding bouts of immobility of 60 s and greater, which are highly correlated with increased arousal threshold, one of the hallmarks of sleep (*Yoshizawa et al., 2015*). For drug treatment studies, fish were allowed normal acclimation periods, followed by 24 hr of baseline recording. At ZT0 fish were treated with either control dimethyl sulfoxide solvent (0.1% DMSO) or freshly prepared TCSOX229 (Tocris), EMPA (Tocris), Suvorexant (Adooq), or YNT-185 (Adooq) (*Hoyer et al., 2013*; *Kummangal et al., 2013*; *Malherbe et al., 2009*; *Plaza-Zabala et al., 2012*) diluted to a final concentration of 1–30 µM into each recording chamber and behavior was recorded for 24 hr.

### Injection procedures

The *hcrt* morpholino sequence 5'-TGGGCTTGGTGTGATCACCTGTCAT-3' was designed (Gene Tools, LLC) based on the available sequence ENSAMXG00000000473 (Ensembl). The *hcrt*-MO targets the first 25 bp of the *hcrt* open reading frame (ORF) to block translation via steric hinderance. Control injections were performed using a standard scrambled sequence 5'CCTCTTACCTCAGTTACAATTTATA-3' (Gene Tools, LLC). Embryos were injected at the 1–2 cell with a 1 nl volume using a pulled borosilicate capillary with a PLI-A100 picoinjector (Warner Instruments) for a final concentration of 0.2 mM morpholino, in accordance with previously published methods (*Bilandžija et al., 2013*; *Gross et al., 2009*). Survival of all embryos was monitored every 6 hr for the first 96 hr of development until behavior was recorded over the next 24 hr. *hcrt*-Gal4:UAS-*BoTx-BLC-GFP* injections were carried out as follows: Plasmid DNA for the Gal4 and the UAS were coinjected into 1–4 cell stage embryos using a pulled borosilicate capillary with PLI-A100 picoinjector (Warner Instruments) at concentration of 25 ng/µL. Tol2 mRNA was coinjected in the

cocktail at a concentration of 25 ng/µL. UAS-*BoTx-BLC-GFP* was injected alone at 25 ng/µL to serve as an injected control group for both Surface and Pachón cavefish. All injected fish were raised to 25 dpf in standard conditions, when behavioral recordings were carried out. Brains from all fish recorded for behavior were dissected and processed for immunohistochemistry in order to quantify by GFP the number of cells silenced by expressing hcrt-Gal4:UAS-BoTx-BLC-GFP.

## Vital dye labeling and lateral line ablation

Fish were treated with 0.002% gentamicin sulfate as previously described (Sigma Aldrich, Carlsbad, CA 1405-41-0) (*Van Trump et al., 2010*). Following baseline sleep recording and neuromast imaging, fish were bathed in gentamicin for 24 hr. Following the treatment, a complete water change was administered and behavior was again recorded for 24 hr. Fish treated with gentamicin were housed in separate tanks for at least 1 month after treatment in order to avoid contamination. Lateral line re-growth was measured with DASPEI staining two weeks following ablation to confirm that there were no long-term effects from the ablation treatments.

## Sequence analysis

To compare Hypocretin/Orexin (HCRT), we aligned the accessioned protein sequences of *A. mexicanus* surface fish (SRR639083.116136.2, SRA) and Pachón cavefish (ENSAMXP00000000478) to orthologous HCRT in zebrafish (ENSDARP00000095322), Medaka (ENSORLP00000004866), Tetraodon (ENSTNIP00000014660), mouse (ENSMUSP00000057578) and human (ENSP00000293330). Protein alignment, neighbor joining tree (cladogram) and sequences analyses were performed with Clustal Omega (v.1.2.1, EMBL-EBI, [*Sievers et al., 2011*]). HCRT domains (PF02072/IPR001704) were determined using Ensembl genome browser (v.83, EMBL-EBI/Sanger) and PFam/Interpro (v.28.0, EMBL-EBI).

## Quantitative PCR (qPCR)

To measure levels of *hcrt* mRNA, whole brains of one-two year old fish were extracted immediately after behavior was recorded (ZT2). After extraction, individual brains were frozen and homogenized in trizol (QIAGEN, Valencia,CA). RNA was extracted with an RNeasy Mini Kit (QIAGEN, Valencia, CA). All RNA samples were standardized to 10 ng/µL concentrations and cDNA synthesis was carried out using iScript (BioRad, Redmond, WA). RT-qPCR was carried out using SsoAdvanced Universal SYBR Green Supermix (BioRad, Redmond, WA) with DNase to eliminate genomic contamination (QIAGEN, Valencia, CA). PCR primers were used at an annealing temperature of 53.3°C; their sequences follow: *hcrt* forward 5'-CAT-CTC-CTC-AGC-CAA-GGT-TT-3', *hcrt* reverse 5'-TAG-AGT-CCG-TGC-TGT-TAC-ACC-3'. *Gapdh* was used as the control gene, and was amplified with the following primer set: forward 5'-TGT-GTC-CGT-GGT-GGA-TCT-TA −3', reverse 5'-TGT-CGC-CAA-TGA-AGT-CAG-AG-3'. Primers were designed to span an exon boundary to eliminate potential for genomic contamination and a melting curve analysis was performed post amplification to confirm a single amplicon for each target gene. The following qPCR protocol was run on a Bio Rad CFX96 with a C1000 thermal cycler: 95.0°C for 3 min followed by a plate read at 95.0°C for 10 s to 53.3°C for 30 s followed by a plate read repeated 39 times. qPCR efficiency was calculated to be 100% as reported by CFX Manager software 3.1, and melt curve analysis showed a single peak for each gene used in this study. All samples were compiled into Bio Rad CFX manager gene study (version 3.1) to account for inter-run calibration. All samples were normalized to one (relative to surface fish controls).

## Immunohistochemistry

Following euthanasia in MS-222 (Sigma Aldrich, Carlsbad, CA) and ice-water, brains were immediately dissected from adults in ice-cold PBS and fixed overnight in 4% Paraformaldehyde/1x PBS (PFA). Adult brains were then placed in 20% sucrose for cryoprotection overnight or until the brains sunk to the bottom of the well (*Kaslin et al., 2004*). Whole brains were then flash frozen and mounted in OCT compound (23-730-571 Fisher scientific) for sectioning. Whole brains were serial sectioned in 50 µm slices, all slices were floated in PBS to rinse out embedding solution. Slice sections were then washed in 0.5% Triton-X 100/PBS (PBT) for 3 × 15 min and co-incubated in 0.5% PBT and 2% Bovine serum albumin (BSA) (Sigma) with primary antibody anti-ORX-A 1:2000 (RRID:

AB_2117647 EMD Millipore) overnight at 4°C in. The slices were rinsed again in 0.5% PBT, 3X for 15 min and placed in secondary antibody 1:600 (Goat anti-rabbit 488; Life Technologies) for 90 min at room temperature. Slices were mounted on slides in Vectashield with DAPI (VectorLabs) and imaged on a Nikon A1 confocal microscope. Whole-mount larvae were fixed overnight in 4% PFA, rinsed 3 × 15 min in 0.5% PBT, then placed in 0.5% PBT with 2% BSA and primary antibody anti-ORX-A 1:2000 overnight. Following 3 × 15 min 0.5% PBT rinse, larvae were placed in secondary antibody 1:600. Larvae were then placed in Vectashield with DAPI until mounted in 2% low melt temp agarose (Sigma Aldrich, Carlsbad, CA) for imaging. All samples were imaged in 2 μm sections and are presented as the Z-stack projection through the entire brain. For quantification of HCRT levels, all hypothalamic slices were imaged in 2 μm sections, merged into a single Z-stack as maximum fluorescence, and the total brain fluorescence was determined by creating individual ROIs for each cell expressing HCRT.

## Statistics

Two-way ANOVA tests were carried out to test the effects of pharmacological and starvation paradigms among different groups and populations on behavior. Each was modeled as a function of genotype (Surface and Pachón) and genotype by treatment interaction (TCS, gentamicin, or starvation, respectively). Significance for all tests was set at $p < 0.05$. When the ANOVA test detected significance, the Holm-Sidak multiple comparison post-test was carried out to correct for the number of comparisons. For comparison of two baseline groups, non-parametric t-tests were carried out to test for significance. Each experiment was repeated independently at least three times. All replicates were biological replicates run independently from one another. No data was excluded, and no statistical outliers were removed. All statistical analysis were carried out using SPSS (IBM, 22.0) or InStat software (GraphPad 6.0). Power analyses were performed to ensure that we had sufficient N to detect significant differences at a minimum of 80% power at the 0.05 threshold using Graphpad InStat.

## Acknowledgements

This work was funded by National Science Foundation Award IOS-125762 to ACK. The authors are grateful to Masato Yoshizawa (Hawai'i) for technical guidance and the Department of Comparative Medicine at FAU for support maintaining the fish facility.

## Additional information

### Funding

| Funder | Grant reference number | Author |
|---|---|---|
| National Science Foundation | Award IOS-125762 | Alex C Keene |

The funders had no role in study design, data collection and interpretation, or the decision to submit the work for publication.

### Author contributions

James B Jaggard, Conceptualization, Data curation, Formal analysis, Investigation, Methodology, Writing—original draft, Writing—review and editing; Bethany A Stahl, Conceptualization, Formal analysis, Investigation, Writing—original draft, Writing—review and editing; Evan Lloyd, Formal analysis, Investigation, Methodology, Writing—original draft, Writing—review and editing; David A Prober, Conceptualization, Resources, Validation, Writing—original draft, Writing—review and editing; Erik R Duboue, Conceptualization, Resources, Software, Formal analysis, Visualization, Methodology, Writing—original draft, Writing—review and editing; Alex C Keene, Conceptualization, Supervision, Funding acquisition, Visualization, Writing—original draft, Project administration, Writing—review and editing

## Author ORCIDs

James B Jaggard ⓘ http://orcid.org/0000-0003-4091-9946
Bethany A Stahl ⓘ http://orcid.org/0000-0001-9218-2996
David A Prober ⓘ http://orcid.org/0000-0002-7371-4675
Erik R Duboue ⓘ http://orcid.org/0000-0003-3303-5149
Alex C Keene ⓘ http://orcid.org/0000-0001-6118-5537

## Ethics

Animal experimentation: This study was performed in strict accordance with the recommendations in the Guide for the Care and Use of Laboratory Animals of the National Institutes of Health. All of the animals were handled according to approved institutional animal care and use committee (IACUC) protocols (#A15-32 and A16-04) of Florida Atlantic University. The protocol was approved by the Committee on the Ethics of Animal Experiments of Florida Atlantic University (#A15-32 and A16-04), and all experiments were performed with oversight from FAU's Department of Animal Medicine.

## Decision letter and Author response

Decision letter https://doi.org/10.7554/eLife.32637.030
Author response https://doi.org/10.7554/eLife.32637.031

## Additional files

### Supplementary files

• Transparent reporting form
DOI: https://doi.org/10.7554/eLife.32637.028

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
