## [Decision Letter]

Thank you for submitting your work entitled "Hypocretin underlies the evolution of sleep loss in the Mexican cavefish" for consideration by *eLife*. Your article has been reviewed by three peer reviewers, and the evaluation has been overseen by a Senior Editor. Our decision has been reached after extensive consultation between the reviewers. Based on these discussions and the individual reviews below, we regret to inform you that your work will not be considered further for publication in *eLife* in its present form.

While all three reviewers found the work to potentially of great interest, they also raised significant concerns. In particular, the reviewers would like to see addition of loss of function experiments to confirm the role of hypocretin. We would be open in the future to a new submission that addresses the concerns of the reviewers, such that every effort would be made to return the paper to the original reviewers. We hope you find the comments of the reviewers, appended below, helpful in revising the manuscript.

*Reviewer #1:*

The manuscript "Hypocretin underlies the evolution of sleep loss in the Mexican cavefish" by Jaggard et al. addresses the neurological basis of sleep loss in cavefish. The authors found that the number of hypocretin neurons is increased in cavefish correlating with the previously observed sleep loss in cavefish populations. The authors use pharmacological treatments to dissect the neurological basis of the sleep phenotype in cave and surface fish. In addition, the authors link the changes in hypocretin activity to physiological phenotypes in the cavefish suggesting not only a plastic behavioral response but also plasticity in the underlying neuronal workings of hypocretin. The study is interesting and carefully executed. I fully support publication in *eLife*. I have only two comments, one technical, one conceptual.

I have some slight concerns with the quantitative PCR data. For example, I was not able to find the primer sequences for rpl13α in the publicly available genome information. In the case for *gapdh* the primers are designed poorly as they span only a very small intron and would give a similarly sized product from genomic DNA which is a common contamination in cDNA preparations no matter how careful the cDNA is prepared. Given that the qPCR data is crucial for the message of the manuscript, I would like to see some better designed controls.

While the plasticity part is quite exciting, I have some trouble to fully comprehend its implications. I can see that starvation is triggering a reduction in wakefulness, presumably through hypocretin. I am not sure, however, how novel or important this is for this study and how this compares to other systems? Also, how can the authors distinguish between causality and it being simply a readout of sleep need? Same is true for the lateral line part, what is the mechanistic basis of the reduction in lateral line signaling leading to a reduction in hypocretin signaling and how does this correlate with the presumably developmental differences that the authors mention in the Discussion (first paragraph)? I feel the manuscript would profit from an extended explanation of this part.

*Reviewer #2:*

This is a very important manuscript that provides a new link between the wakefulness promoting neuropeptide hypocretin and previously discovered differences in sleep behavior in surface- and cave-dwelling *Astyanax mexicanus*. The major finding is that upregulation of hypocretin at both the transcript and protein levels is correlated with increased wakefulness in cavefish. Pharmacological data support a direct relationship between the neuropeptide function and the cavefish behavior. Because hypocretin neurons are found in the hypothalamus this discovery provides a possible explanation for why this part of the brain is modified during the evolution of cavefish. The novel information within this manuscript will markedly improve our general understanding of the conserved role of hypocretin in the evolution of sleep regulation.

The following comments are offered to improve the manuscript.

The results show that knocking down hypocretin with a pharmacological inhibitor increases sleep duration in cavefish. However, there is no effect on surface fish. Could an effect on surface fish be achieved by knocking up hypocretin, possibly by injecting transcripts, an expression construct containing the gene, or the protein?

To extend the above comment, there is no explanation given as to why pharmacological inhibition of hypocretin does not affect surface fish if this neuropeptide is indeed a general regulator of sleep behavior.

Although controls for the quantitative PCR analyses are mentioned in the Materials and methods, I cannot find a place where the control data is illustrated in the manuscript. Ideally, the control data for quantitative PCR should be presented for evaluation alongside the experimental results. I understand that this may be a part of the "normalization" process, but it is stated that normalization was relative to surface fish controls (subsection “Quantitative PCR (qPCR)”). Perhaps the "control" genes themselves change between surface fish and cavefish. More information is needed about this possibility.

Why are the critical results for gentamycin treatment printed so lightly in Figure 3, such that even the orbits are difficult to distinguish? Are there no neuromasts at all that survive the treatment? More contrast needs to be included in the photographs for the reader to be convinced about the effects of the treatment on the levels of DASPEI stained neuromasts. This is particularly true for cavefish, in which superficial neuromasts in the head are more numerous than in surface fish.

The quantitative results shown in Figure 3 do not seem to correspond well to the representative photographs in Figure 3. It seems like there are more hypocretin stained cells in Figure 3 than in K, not only a decrease in the level of staining in each cell, and this is not evident in the quantification in Figure 3. The same is true for Figure 4, and G. Perhaps the images need to be enlarged.

To continue with the comment immediately above, it appears that there are variable levels of hypocretin staining in cell clusters in different parts of the section. Is this real, and if so, does it correspond to particular centers within the hypothalamus?

In the Discussion, it states that there are "no differences in the (sic hypocretin) genomic sequence" between the two morphs. A correction is needed because only exon sequences could have been determined from the transcriptome data of Gross et al. (2013).

In the fourth paragraph of the Discussion, the authors point out that the wake-promoting role of HCRT neurons is dependent on norepinephrine signaling. However, they fail to point out that norepinephrine has been demonstrated to be increased in cavefish (see Bilandzija et al., 2013, PloS ONE 8 (11): e80823). This information is important here. It should be added and the Discussion revised accordingly.

*Reviewer #3:*

Jaggard et al. examines hypocretin neurons in two different populations of the Mexican cavefish, *Astyanax mexicanus*. They identified difference in hypocretin neuron cell numbers in the surface-dwelling vs cave-dwelling populations. Pharmacological inhibition of HCRT signaling increases sleep duration in cavefish. Further they perform manipulations that promote sleep, such as lateral line ablation and starvation, and show that hcrt expression is inhibited in cavefish and not in surface fish. Based on these results the authors conclude that alteration in HCRT signaling contribute to the evolution of sleep loss.

Given the conserved role of HCRT in sleep regulation in different species, the conclusion of this study seems logical and expected. However, their experimental results are insufficient to support this conclusion. The difference in hypocretin neuron cells numbers between these two species are clear, yet beyond this correlation the evidence that the study provides attempting to causally link HCRT and sleep loss is weak.

1) The authors draw their conclusion based on three experimental manipulations- pharmacological manipulation of Hcrt receptor 2, lateral line ablation using gentamicin, and starvation. The only "specific" manipulation for Hcrt system is the pharmacological manipulation using TCSOX229 (TCS). The authors cite Plaza-Zabala et al. 2012, which shows the application of this HCRTR2 in mice. A big issue in this study is that there is no control to show that the application actually affects hypocretin neurons in zebrafish. What is the half-life of TCS in water? How do we know that TCS is selective for HCRTR2 in zebrafish? The protein sequences of HCRTR2 in zebrafish and mouse are different and what is the evidence that TCS is a specific HCRTR2 antagonist in zebrafish? Even if this has been shown elsewhere, the authors need to demonstrate that it works in their lab in the context of this study as well. For example, the authors could show altered activity of cells expressing HCRTR upon TCS treatment using c-fos, p-ERK staining, etc. Also how is the dosage determined? What is the evidence that TCS actually penetrates and reaches the hypothalamic hypocretin neurons when supplied in bath water at this concentration? In mice TCS is administered by IP route in a volume of 5 mL/kg body weight. In this study the TCS doses were based on a pilot experiment testing the effects of this antagonist on the hyperactivity induced by hypocrein-2 in C57BL/6J mice. A similar experiment using hypocretin overexpression or knock out could be done to determine the effective concentration.

2) Pharmacological manipulations can have pleiotropic effects depending on the dosage even if the authors show that it targets HCRTR2. The possibility that it can target other proteins in addition to HCRT will be difficult to rule out. Therefore the authors need to show independent lines of evidence that HCRT is involved in sleep loss using genetic manipulation using morpholinos against HCRT/HCRTR and overexpressing HCRT/HCRTR, which should show the opposite phenotype from morpholino treatment.

3) Presumably many other cell types other than HCRT could show differences between surface and cave-dwelling types apart from HCRT. Modulation of several neuropeptide/neurotransmitter systems could affect sleep dramatically. For example, differences in CRH (corticotropin-releasing- hormone) or histaminergic, or NPY system could alter sleep. The authors should examine differences in other major neuropeptide systems that are involved in sleep regulation between surface and cave fish.

4) I am confused about the some of the supplementary figure information. For example, what is the difference between Figure 2A, B vs. Figure 2—figure supplement 1A and B. Are they both treated with TCS as the figures themselves indicate?

[Editors’ note: what now follows is the decision letter after the authors submitted for further consideration.]

Congratulations, we are pleased to inform you that your article, "Hypocretin underlies the evolution of sleep loss in the Mexican cavefish", has been accepted for publication in *eLife*.

The authors investigate the role of a hypocretin in controlling sleep in the blind Mexican cavefish. The results suggest that evolutionary changes in hypocretin regulation may have been a driving factor in the evolution of sleep loss. On further discussion the reviewers agreed that the morpholino controls recommended by reviewer #2 (below) are not essential for the publication of this work.

*Reviewer #1:*

In the revised manuscript "Hypocretin underlies the evolution of sleep loss in the Mexican cavefish" by Jaggard et al. the authors have addressed all my previous comments. The manuscript is now substantially improved and provides several novel technical advances in the cavefish field. The use of Gal4 UAS in mosaic transgenics is elegant and very convincing. The morpholino results by its own would technically require some additional controls (e.g. RT-PCR of splice morpholino or RNA rescue), however, as they only serve as additional evidence, their use strengthen the other results significantly, especially as the authors have established their behavioral assays now in larval fish which will be useful for future studies.

There are still some oversight mistakes but I assume these can be taken care through editorial revision (e.g. Results, fifth, sixth, seventh and ninth paragraphs; Discussion second and third paragraphs).

*Reviewer #2:*

This manuscript is the resubmission of a previous manuscript submitted to *eLife*, that was not accepted for publication. The major criticism was that not enough evidence, above pharmacological studies, was included to strongly support the possibility that hypocretin is evolved in cavefish sleep loss, and that it was unlikely that the required information could be produced within the time frame required by the journal. The present manuscript answers some (but not all) of the previous objections, and is considerably improved over the previous submission.

An improvement is the development of a morpholino-based procedure for knockdown of the hypocretin gene, which resulted in an increase in sleep duration in cavefish, without affecting sleep in surface fish, as predicted by the authors’ hypothesis.

Morpholinos have been in use for some time in cavefish, most recently by Bilandzija (2013) for oca2 knockdown. In the latter study, it was simple to determine the morpholino effects because visible pigmentation was reduced or eliminated. In other morpholino studies, however, controls in addition to scrambled morpholinos were done to be certain the that morpholinos were working on the expected targets. Such controls are not reported here, and would be seem to be necessary for publication in *eLife*.

A translation blocking morpholino is used in these experiments, so the appropriate control would be to compare hypocretin protein levels using the available antibody, either in Western blots (preferred for more exact quantification) or in immuno-imaging experiments of the brain. These controls could be done rather quickly.

*Reviewer #3:*

In this manuscript, Jaggard et al. investigate HCRT and its role in controlling sleep in *Astyanix mexicanus*, the blind Mexican cavefish. Through pharmacological and genetic manipulations, the authors show that an increase in HCRT is associated with sleep loss in the Pachón cavefish. Comparing the results of the cave population to the surface population, along with previous zebrafish and mouse data, the authors conclude that evolutionary changes in HCRT function and regulation may have been a driving factor in the evolution of sleep loss in the Pachón cavefish.

This is a very interesting paper. The authors addressed all of the reviewer's concerns and the data presented will, no doubt, further understanding of HCRT and its evolutionary role in sleep regulation as well as provide fuel for further experiments addressing the evolution of sleep.

The data presented is very thorough and the authors' interpretation of the role of HCRT in sleep regulation of the Pachón cavefish population is very convincing.

---

## [Author Response]

Reviewer #1:

[…] I have some slight concerns with the quantitative PCR data. For example, I was not able to find the primer sequences for rpl13α in the publicly available genome information. In the case for gapdh the primers are designed poorly as they span only a very small intron and would give a similarly sized product from genomic DNA which is a common contamination in cDNA preparations no matter how careful the cDNA is prepared. Given that the qPCR data is crucial for the message of the manuscript, I would like to see some better designed controls.

We originally chose to standardize qPCR experiments against *rpl13a* and *gapdh* because these had been used in resent *A. mexicanus* papers (Beale et al., 2013, Stahl and Gross, 2015). In the original submission there was a typo in the reported primer sequences for *gapdh,* which has been corrected in the revised manuscript. These specific primers have been used repeatedly with success in multiple publications (Gross and Wilkens 2013, Stahl and Gross 2015, and Gross et al. 2016). However as requested, we re-analyzed and validate that these primers are appropriately designed for our purposes.

These *gapdh* primers amplify a cDNA product of 152bp, which spans two exons. In the scenario that contaminated gDNA were to be amplified, this would yield a 300bp amplicon. To confirm amplification of our specific cDNA product, we performed a melt curve analysis post qPCR amplification, which has been widely used to detect small differences in amplicon size for genotyping (Herrmann et al. 2006) and can easily resolve out peaks for products differing 150bp in length (Ririe KM, Rasmussen, P et al. 1997). Consistent with standard qPCR guidelines (Taylor S, Wakem M, et al. 2010), our melt curve analysis yielded a single peak for *gapdh*, therefore we determined these primers are consistent and sufficient for our analyses.

Further, we have reported the values of housekeeping genes in (Figure 1—figure supplement 1) and report the M-values for housekeeping efficiency, along with additional details described above to the Materials and methods section.

At the reviewer’s suggestion we have reanalyzed the targets of the previously published *rpl13a* primers (Beale et al. 2013). These were originally selected based on being ‘remarkably consistent between all *A. mexicanus* populations (Beale et al., 2013). However, publication of the cavefish genome (McGaugh et al. 2014) and recent re-sequencing releases available on NCBI do not reveal targets for these, and they instead match perfectly to the *Danio* genome. We believe this is most likely due to sequencing errors in the *A. mexicanus* genome. However, out of an abundance of caution, we have omitted *rpl13a* from the revised manuscript. All qPCR data in the resubmission standardize against *gadph* alone.

While the plasticity part is quite exciting, I have some trouble to fully comprehend its implications. I can see that starvation is triggering a reduction in wakefulness, presumably through hypocretin. I am not sure, however, how novel or important this is for this study and how this compares to other systems? Also, how can the authors distinguish between causality and it being simply a readout of sleep need? Same is true for the lateral line part, what is the mechanistic basis of the reduction in lateral line signaling leading to a reduction in hypocretin signaling and how does this correlate with the presumably developmental differences that the authors mention in the Discussion (first paragraph)? I feel the manuscript would profit from an extended explanation of this part.

The current version describes work in fish and mice indicating that Hcrt is regulated by sensory stimuli and by leptin regulation. We propose that Hcrt is critical for sleep plasticity in response to environmental perturbations. With respect to causality, this is difficult to directly address, though it is widely believed that Hcrt is not an output of sleep regulation. With respect to causality, we hope that experiments inhibiting Hcrt function (pharmacological, genetic silencing, and morpholino) now support the notion that inhibiting Hcrt function in cavefish promotes sleep.

The reviewer also noted a number of typos/grammatical errors. We thank the reviewer for noticing these minor errors and hope to have fixed these in the revised manuscript.

Reviewer #2:

[…] The following comments are offered to improve the manuscript.The results show that knocking down hypocretin with a pharmacological inhibitor increases sleep duration in cavefish. However, there is no effect on surface fish. Could an effect on surface fish be achieved by knocking up hypocretin, possibly by injecting transcripts, an expression construct containing the gene, or the protein?

The current version includes numerous additional experiments to support initial pharmacological findings. First, we show similar effects to those first reported with the HCRTR2 antagonist TCS with two additional antagonists (Suvorexant and EMPA; Figure 2). Second, we show that an Hcrt agonist (YNT-185) reduces sleep in surface fish (Figure 2—figure supplement 1). Third, we show that morpholino treatment restores sleep to 4 dpf Pachon cavefish (Figure 4). Fourth, we use mosaic silencing of Hcrt neurons through transgenic expression of botulinum toxin restores sleep in cavefish (Figure 5).

To extend the above comment, there is no explanation given as to why pharmacological inhibition of hypocretin does not affect surface fish if this neuropeptide is indeed a general regulator of sleep behavior.

We have clarified that the results suggest cavefish sleep is more sensitive to inhibition of Hcrt signaling and a phenotype with drug treatment is not observed in surface fish at the concentrations used, rather than sleep being Hcrt independent in surface fish (see Discussion, second paragraph). Specifically, we state ‘All three manipulations employed are likely to only partially disrupt HCRTR signaling, and it is likely that complete inhibition of HCRTR signaling would increase sleep in surface fish.’

Although controls for the quantitative PCR analyses are mentioned in the Materials and methods, I cannot find a place where the control data is illustrated in the manuscript. Ideally, the control data for quantitative PCR should be presented for evaluation alongside the experimental results. I understand that this may be a part of the "normalization" process, but it is stated that normalization was relative to surface fish controls (subsection “Quantitative PCR (qPCR)”). Perhaps the "control" genes themselves change between surface fish and cavefish. More information is needed about this possibility.

We have added the control data to Figure 1—figure supplement 1. While both primer sets have previously been published in *A. mexicanus*, the current manuscript exclusively standardizes against *gadph* (Beale et al. 2013; Stahl and Gross 2015, see response to reviewer #1). Additionally, we have reported the M-values in the Materials and methods section, both of which fall well within the desirable range for quality housekeeping genes.

Why are the critical results for gentamycin treatment printed so lightly in Figure 3, such that even the orbits are difficult to distinguish? Are there no neuromasts at all that survive the treatment? More contrast needs to be included in the photographs for the reader to be convinced about the effects of the treatment on the levels of DASPEI stained neuromasts. This is particularly true for cavefish, in which superficial neuromasts in the head are more numerous than in surface fish.

We have enhanced the contrast of the images in question to demonstrate that indeed the entire neuromast system is ablated with this concentration of gentamicin treatment. These findings are also supported by previous literature (Van Trump et al. 2010; Yoshizawa et al. 2010, 2013; Jaggard et al. 2017) indicating complete ablation of neurons with gentamicin treatment.

The quantitative results shown in Figure 3 do not seem to correspond well to the representative photographs in Figure 3. It seems like there are more hypocretin stained cells in Figure 3 than in K, not only a decrease in the level of staining in each cell, and this is not evident in the quantification in Figure 3. The same is true for Figure 4, and G. Perhaps the images need to be enlarged.

We believe it appears that there are more cells because of their brightness, and that closer inspection during quantification suggests they are equivalent. It is worth noting that these images represent a single 2 μm optical imaging section in order to better highlight single cells, and therefore does not offer a full view of the entire HCRT system of the hypothalamus, which is in the range of 500 μm in adult fish. We have enlarged the images in question as much as space allows and hope this accurately represents our findings.

To continue with the comment immediately above, it appears that there are variable levels of hypocretin staining in cell clusters in different parts of the section. Is this real, and if so, does it correspond to particular centers within the hypothalamus?

While there do seem to be regions in which antibody labelling is stronger or weaker, we did not see a trend in conserved nuclei expressing these changes among biological replicates, and can most likely be attributed to slight variability among animals and/or staining quality in replications. Moreover, we have quantified fluorescence intensity among individuals, and we present the variability in the error bars in Figure 3 and 4.

In the Discussion, it states that there are "no differences in the (sic hypocretin) genomic sequence" between the two morphs. A correction is needed because only exon sequences could have been determined from the transcriptome data of Gross et al. (2013).

Thank you for noticing this omission. We have now clarified that analysis is limited to exons.

In the fourth paragraph of the Discussion, the authors point out that the wake-promoting role of HCRT neurons is dependent on norepinephrine signaling. However, they fail to point out that norepinephrine has been demonstrated to be increased in cavefish (see Bilandzija et al., 2013, PloS ONE 8 (11): e80823). This information is important here. It should be added and the Discussion revised accordingly.

Thank you for noticing this oversight. Indeed, the Bilandzija study was one of the findings that pointed us towards studying Hcrt and has now been appropriately added to the Discussion.

We have corrected the typos/minor errors in the original version.

Reviewer #3:

[…] Given the conserved role of HCRT in sleep regulation in different species, the conclusion of this study seems logical and expected. However, their experimental results are insufficient to support this conclusion. The difference in hypocretin neuron cells numbers between these two species are clear, yet beyond this correlation the evidence that the study provides attempting to causally link HCRT and sleep loss is weak.1) The authors draw their conclusion based on three experimental manipulations- pharmacological manipulation of Hcrt receptor 2, lateral line ablation using gentamicin, and starvation. The only "specific" manipulation for Hcrt system is the pharmacological manipulation using TCSOX229 (TCS). The authors cite Plaza-Zabala et al. 2012, which shows the application of this HCRTR2 in mice. A big issue in this study is that there is no control to show that the application actually affects hypocretin neurons in zebrafish. What is the half-life of TCS in water? How do we know that TCS is selective for HCRTR2 in zebrafish? The protein sequences of HCRTR2 in zebrafish and mouse are different and what is the evidence that TCS is a specific HCRTR2 antagonist in zebrafish? Even if this has been shown elsewhere, the authors need to demonstrate that it works in their lab in the context of this study as well. For example, the authors could show altered activity of cells expressing HCRTR upon TCS treatment using c-fos, p-ERK staining, etc. Also how is the dosage determined?

We appreciate this suggestion and have now acknowledged the limitation of pharmacological agents in this system (for example, Discussion, second paragraph). We currently do not have the genetic tools to examine Hcrtr2, though we are actively working to generate them. We have also initiated a collaboration (inspired by this suggestion) with Laura Bohn at Scripps to examine the effects of the drugs used in these experiments on zebrafish and *A. mexicanus* HCRTR2 signaling. While we are eager to include these experiments, they are not trivial, even with support from the Bohn lab, and we therefore hope they are not essential. We have instead used additional drugs, genetic approaches, and morpholinos to fortify the conclusions. In particular, the genetic knockdown of *hcrt* and genetic silencing of HCRT neurons promoting sleep demonstrate that the relationship between HCRT signaling and sleep differences in cave and surface fish are causally linked. While the genetic tools in cavefish are not at the same standard as those in zebrafish, we are highly committed to transferring zebrafish genetic technology to cavefish and are collaborating with a number of zebrafish labs to efficiently implement transgenesis and gene-editing. We hope that future endeavors from our group will include this level of resolution.

What is the evidence that TCS actually penetrates and reaches the hypothalamic hypocretin neurons when supplied in bath water at this concentration? In mice TCS is administered by IP route in a volume of 5 mL/kg body weight. In this study the TCS doses were based on a pilot experiment testing the effects of this antagonist on the hyperactivity induced by hypocrein-2 in C57BL/6J mice. A similar experiment using hypocretin overexpression or knock out could be done to determine the effective concentration.

While the ability of drugs to cross the blood-brain-barrier has not been tested in fish, a number of the HCRTR2 modulators reportedly pen0etrate the brain of rodents. We have now stated our justification for drug selection (Results, fourth paragraph). We initially selected our concentration based on those previously used to study sleep drugs in zebrafish (Rihel et al., 2010) and our studies in cavefish (Duboue 2012). In addition, the drugs used were prohibitively expensive at higher concentrations, particularly for adult studies. We acknowledge that without understanding receptor pharmacology in our system, this is effectively an educated guess, though our approach is based on precedent set in studies in zebrafish. We have also bolstered our findings using complementary approaches.

2) Pharmacological manipulations can have pleiotropic effects depending on the dosage even if the authors show that it targets HCRTR2. The possibility that it can target other proteins in addition to HCRT will be difficult to rule out. Therefore the authors need to show independent lines of evidence that HCRT is involved in sleep loss using genetic manipulation using morpholinos against HCRT/HCRTR and overexpressing HCRT/HCRTR, which should show the opposite phenotype from morpholino treatment.

Thank you for this suggestion. While we have not been able to overexpress Hcrt, we have tested an HCRTR2 agonist that reduces sleep in surface fish (Figure 3—figure supplement 2). In addition, we have found that morpholinos and mosaic silencing of Hct restore sleep to cavefish. We fully recognize that none of these experiments by themselves are optimal. However, both approaches represent a technical advance for our field. For example, the use of morpholinos required measuring sleep in 3-4 dpf fish, which had not been performed. In addition, the UAS-Btx experiments represent the first use of GAL4/UAS in *A. mexicanus*. These experiments have inspired us to generate additional tools (such as Hcrt:GFP and nfsb lines), however, the analysis is not ready to be included in this manuscript.

3) Presumably many other cell types other than HCRT could show differences between surface and cave-dwelling types apart from HCRT. Modulation of several neuropeptide/neurotransmitter systems could affect sleep dramatically. For example, differences in CRH (corticotropin-releasing- hormone) or histaminergic, or NPY system could alter sleep. The authors should examine differences in other major neuropeptide systems that are involved in sleep regulation between surface and cave fish.

We fully agree and have added discussion of this (Discussion, last paragraph). We do not mean to suggest that Hcrt alone regulates sleep loss in this system. Rather, our conclusion is that the evolution of Hcrt function contributes to the evolution of sleep loss in this system. We are addressing this in our current work in a number of ways including a collaboration with the Appelbaum Lab to examine Hcrt-associated circuitry in zebrafish and *A. mexicanus.* Again, these experiments are challenging in both systems, and we are unable to incorporate it into this manuscript. In addition, we are eager to examine the evolution of known sleep genes (in flies, zebrafish, and mice) as well as genes identified through our ongoing QTL mapping efforts. We describe these efforts in the Discussion.

4) I am confused about the some of the supplementary figure information. For example, what is the difference between Figure 2 vs. Figure 2—figure supplement 1A and B. Are they both treated with TCS as the figures themselves indicate?

In the original version, the main panel was in adults, while the supplemental was in fry. In the revised version we have taken care to label and describe figures more clearly.